# IMAGINARYNET: LEARNING OBJECT DETECTORS WITHOUT REAL IMAGES AND ANNOTATIONS

**Minheng Ni, Zitong Huang, Kailai Feng & Wangmeng Zuo**[*]
Faculty of Computing
Harbin Institute of Technology
{mhni, zthuang, klfeng}@stu.hit.edu.cn   wmzuo@hit.edu.cn

## ABSTRACT

Without the demand of training in reality, humans are able of detecting a new category of object simply based on the language description on its visual characteristics. Empowering deep learning with this ability undoubtedly enables the neural network to handle complex vision tasks, *e.g.*, object detection, without collecting and annotating real images. To this end, this paper introduces a novel challenging learning paradigm Imaginary-Supervised Object Detection (ISOD), where neither real images nor manual annotations are allowed for training object detectors. To resolve this challenge, we propose IMAGINARYNET, a framework to synthesize images by combining pretrained language model and text-to-image synthesis model. Given a class label, the language model is used to generate a full description of a scene with a target object, and the text-to-image model is deployed to generate a photo-realistic image. With the synthesized images and class labels, weakly supervised object detection can then be leveraged to accomplish ISOD. By gradually introducing real images and manual annotations, IMAGINARYNET can collaborate with other supervision settings to further boost detection performance. Experiments show that IMAGINARYNET can (i) obtain about 75% performance in ISOD compared with the weakly supervised counterpart of the same backbone trained on real data, (ii) significantly improve the baseline while achieving state-of-the-art or comparable performance by incorporating IMAGINARYNET with other supervision settings. Our code will be publicly available at https://github.com/kodenii/ImaginaryNet.

## 1 INTRODUCTION

Without the demand of training in reality, humans are able of detecting a new category of object simply based on the language description on its visual characteristics. Equipping this ability to deep learning may allow the neural network to handle complex vision tasks, *e.g.*, object detection, without real images and annotations. Recently, we witness the rise of Contrastive Language-Image Pre-training (CLIP) (Radford et al., 2021), where general knowledge can be learned by pre-training and then be applied to various downstream tasks via zero-shot learning or task-specific fine-tuning. Unlike image classification, object detection is more challenging and has a larger gap than pre-training tasks. Several methods, such as RegionCLIP (Zhong et al., 2022a), ViLD (Gu et al., 2021), and Detic (Zhou et al., 2022), have been suggested to transfer knowledge from pre-trained CLIP (Radford et al., 2021) to some modules of detection. However, real images and annotations are still required for some key modules of the object detectors, such as Region Proposal Network (RPN) or Region of Interest (RoI) heads.

In this work, we aim to raise and answer a question: given suitable pre-trained models, can we learn object detectors without real images and manual annotations? To this end, we introduce a novel learning paradigm, *i.e.*, Imaginary-Supervised Object Detection (ISOD), where no real images and manual annotations can be used for training object detection. Fortunately, benefited from the progress in vision-language pre-training, ISOD is practically feasible. Here we propose IMAGINARYNET, a framework to learn object detectors by combining pretrained language model as well

---

[*]Corresponding Author.

as text-to-image synthesis models. In particular, the text-to-image synthesis model is adopted to generate photo-realistic images, and the language model can be used to improve the diversity and provide class labels for the synthesized images. Then, ISOD can be conducted by leveraging weakly supervised object detection (WSOD) algorithms on the synthesized images with class labels to learn object detectors. We set up a strong CLIP-based model as the baseline to verify the effectiveness of IMAGINARYNET. Experiments show that IMAGINARYNET can outperform the CLIP-based model with a large margin. Moreover, IMAGINARYNET obtains about 75% performance in ISOD compared with the weakly supervised model of the same backbone trained on real data, clearly showing the feasibility of learning object detection without any real images and manual annotations.

By gradually introducing real images and manual annotations, IMAGINARYNET can collaborate with other supervision settings to further boost detection performance. It is worthy noting that the performance of existing object detection models may be constrained by the limited amount of training data. As a result, we use IMAGINARYNET as a dataset expansion approach to incorporate with real images and manual annotations. Further experiments show that IMAGINARYNET significantly improves the performance of the baselines while achieving state-of-the-art or comparable performance in the supervision setting.

To sum up, the contributions of this work are as follows:

- We propose IMAGINARYNET, a framework to generate synthesized images as well as supervision information for training object detector. To the best of our knowledge, we are among the first work to train deep object detectors solely based on synthesized images.
- We propose a novel paradigm of object detection, Imaginary-Supervised Object Detection (ISOD), where no real images and annotations can be used for training object detectors. We set up the benchmark of ISOD and obtain about 75% performance in ISOD when compared with the WSOD model of the same backbone trained on real data.
- By incorporating with real images and manual annotations, ImaginaryNet significantly improves the baseline model while achieving state-of-the-art or comparable performance.

## 2 RELATED WORK

### 2.1 OBJECT DETECTION

Most fully-supervised object detection methods (FSOD) (Ren et al., 2015; Redmon et al., 2016; Tian et al., 2019; Carion et al., 2020) rely on large amount of training data with box-level annotations. To reduce the annotation costs, some works attempt to train a detector with incompletely supervised training data. For example, weakly-supervised object detection (WSOD) (Huang et al., 2022; Dong et al., 2021; Bilen & Vedaldi, 2016; Tang et al., 2017) requires only image-level labels. While semi-supervised object detection (SSOD) (Liu et al., 2021; Xu et al., 2021; Chen et al., 2022) leverages unlabeled data combining with box-level labeled data. Although these works use relatively less or weak supervision, all of these works still rely on real images and manual annotations. In this paper, we propose ISOD, where no real images and manual annotations can be used for training object detection, thereby saving the demand for data acquisition and annotation costs.

### 2.2 SIM2REAL

Sim2real aims to use the simulator engines to simulate images for the model. Some works (Akhyani et al., 2022; Sadeghi & Levine, 2016; Wang et al., 2018) use engines to simulate data for different vision tasks, such as emotion classification. In object detection, Horváth et al. (2022); Borrego et al. (2018); Tremblay et al. (2018) use simulated images with 2D or 3D engine, e.g., UE4, for industrial object detection. Because of the large domain gap caused by limitation of simulating engine, Sim2real and subsequent domain adaption methods focus on reducing domain gap. However, with the progress in text-to-image synthesis, domain gap of images generated by such models has been largely reduced. The key problem is how to effectively employ text-to-image synthesis model for generating diverse images with proper content and quality. To this end, we propose IMAGINARYNET, which can even archive 75% performance in ISOD compared with the weakly supervised counterpart of the same backbone.

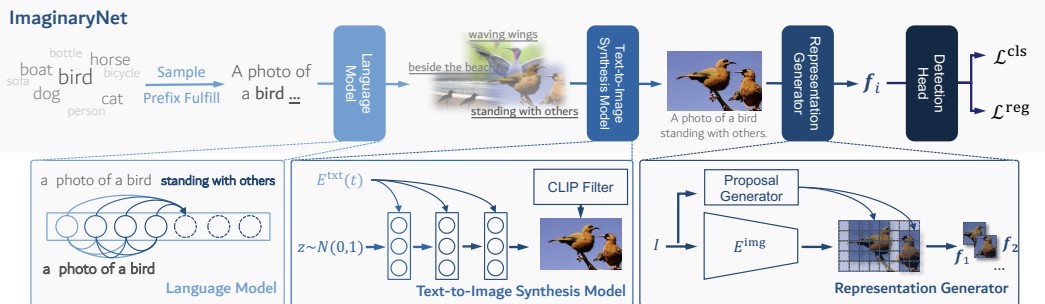

Figure 1: **Overview of IMAGINARYNET.** IMAGINARYNET samples the class label randomly and fulfills it to the prefix template. The language model extends the prefix to a complete description. The synthesis model generates imaginary images from random noise based on the description. Proposal representations are extracted from imaginary images. IMAGINARYNET optimizes Detection Head with proposal representations and class labels. If real data exists, Detection Head will also be optimized based on representations from real images and manual annotations.

## 2.3 PRE-TRAINED MODELS

Pre-trained models have shown effectiveness in many vision tasks including object detection. Some CLIP-based models can transfer knowledge from pre-trained CLIP (Radford et al., 2021) model. However, real images and manual annotations are still needed for their RPNs or RoI heads. Visual synthesis models, such as GAN (Goodfellow et al., 2016), StyleGAN (Karras et al., 2019), and ImageBART Esser et al. (2021) aim to generate plausible images. In recent years, we witness the rise of text-to-image synthesis models with high quality and capability of language controlling, such as DALL-E (Ramesh et al., 2021), Stable Diffusion (Rombach et al., 2022), and Imagen (Saharia et al., 2022). However, few studies have been given to only using the knowledge from pre-trained text-to-image synthesis models to handle complex vision tasks, *e.g.*, object detection.

## 3 METHODOLOGY

### 3.1 PRELIMINARIES AND PROBLEM FORMULATIONS

Object detection aims to find each object with the bounding box $\mathbf{b}_k \in \mathbb{R}^4$, defined by $[x_{\min}, y_{\min}, x_{\max}, y_{\max}]$ that specifies its top-left corner $(x_{\min}, y_{\min})$ and its bottom-right corner $(x_{\max}, y_{\max})$, and class label $c \in \mathcal{C}$ in a given image $\mathbf{I} \in \mathbb{R}^{C \times W \times H}$, where $C$, $W$ and $H$ denote the channels, width and height of the image and $k$ denotes the $k$-th object, respectively.

In WSOD, we have a real dataset $\mathcal{D}_{\mathrm{w}}^{\mathrm{R}} = \{(\mathbf{I}_{\mathrm{w}}^{\mathrm{R}}, \mathbf{Y}_{\mathrm{w}}^{\mathrm{R}})\}$ to train the detector, where $\mathbf{I}_{\mathrm{w}}^{\mathrm{R}}$ is the real image and $\mathbf{Y}_{\mathrm{w}}^{\mathrm{R}} = \left[y_1^{\mathrm{R}}, ..., y_{|\mathcal{C}|}^{\mathrm{R}}\right]$ is the corresponding image-level weak-annotation. Class-labeled annotation $y_c^{\mathrm{R}} \in \{0, 1\}$ indicates the presence of at least one instance of $c$-th category and $|\mathcal{C}|$ is the number of categories. In SSOD, we have a real labeled dataset $\mathcal{D}_{\mathrm{f}}^{\mathrm{R}} = \{(\mathbf{I}_{\mathrm{f}}^{\mathrm{R}}, \mathbf{Y}_{\mathrm{f}}^{\mathrm{R}})\}$ and an un-labeled dataset $\mathcal{D}_{\mathrm{u}}^{\mathrm{R}} = \{(\mathbf{I}_{\mathrm{u}}^{\mathrm{R}})\}$. The real image $\mathbf{I}_{\mathrm{f}}^R$, corresponding instance-level annotations $\mathbf{Y}_{\mathrm{f}}^{\mathrm{R}} = [(\mathbf{b}_1, c_1), \ldots]$ with $N^R$ elements, and unlabeled image $\mathbf{I}_{\mathrm{u}}^R$ can be accessed. Here, $N^R$ is the number of object instances associated with $\mathbf{I}_{\mathrm{f}}^{\mathrm{R}}$, and $\mathbf{b}_i$ is the $i$-th object localization bounding box.

In this paper, we propose Imaginary-Supervised Object Detection (ISOD), where no real images and annotations can be accessed.

### 3.2 IMAGINARY GENERATOR

IMAGINARYNET aims to generate synthesized images as well as supervision information for training object detectors. We divide this into a series steps: 1. Prompt Generation, 2. Image Generation, and 3. CLIP Filtering.

**Prompt Generation** Given a class vocab $\mathcal{C}$, IMAGINARYNET samples uniformly to get a class label $c \in \mathcal{C}$. Then, the class name of $c$ will be fulfilled into a prefix phrase, which is "*A photo of*

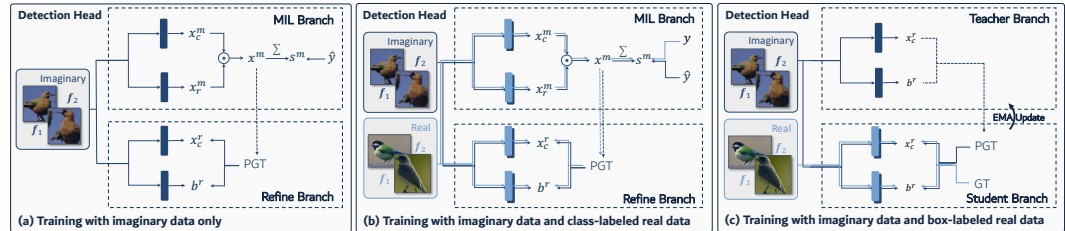

Figure 2: **The structure of Detection Head.** Based on different data settings, we use three types of detection heads. In (a) type of Detection Head, no real images and annotations will participate the training process. In types (b) and (c), real images and annotations will be trained with imaginary data together. In type (c), MIL Branch will be disabled due to box supervision exists.

*a*", to obtain the subject description $P_c$ of the imaginary image. To make the scene of image more specific and diverse, $P_c$ will be the guidance prefix of the language model $G^L$, *e.g.*, GPT-2 (Radford et al., 2019), to imagine the full description of the scene of image. The language model $G^L$, which is a transformer-based autoregressive network, generates the full description token by token until meeting a full stop or exceeding the maximum of length $M^L$ [1]. Let $\mathbf{t}_c = G^L(P_c)$ be the generated full description of the imaginary image.

**Image Generation**    IMAGINARYNET further obtains the synthesized image by

$$\mathbf{I} = G^V(z | \mathbf{t}_c), \tag{1}$$

where $z \sim \mathcal{N}(0, 1)$ is random noise sampled from normal distribution. The text-to-image synthesis model $G^V$ usually adopts a transformer-based autoregressive model or diffusion network, and generates imaginary image $\mathbf{I}$ based on $z$ with the guidance of $\mathbf{t}_c$ in a step by step manner.

**CLIP Filtering**    The language and text-to-image synthesis model may bring in errors during their generation process. To tackle this problem, following the method of CLIP classification (Radford et al., 2021), the CLIP Filter will find out the most matched text in the group $\{P_c \mid c \in \mathbb{C}\}$, e.g., {"A photo of aeroplane", "A photo of bicycle", "A photo of bird", …}, with each image to do classification. If the class of the image classified by CLIP is different from its initial class used for generation, we consider this an error sample and re-generate it from **Prompt Generation**.

### 3.3   DETECTION NETWORK

**Representation Generator**    For a real image or an imaginary image $\mathbf{I}$, IMAGINARYNET obtains its feature map $\mathbf{h} = E^I(\mathbf{I})$, where $E^I$ is the image encoder consisting of multi-layers of convolutional networks. Simultaneously, proposals $\mathcal{P} = R(\mathbf{I})$ of the image $\mathbf{I}$ are extracted by proposal generator $R$, which adopts a selective search module for ISOD and WSOD or an RPN without initialization for SSOD. IMAGINARYNET applies RoI Pooling (Ren et al., 2015; Girshick, 2015) to each proposal $p_i \in \mathcal{P}$ in feature map $h$ to obtain the proposal representation $\mathbf{f}_i \in \mathbb{R}^d$. Consequently, IMAGINARYNET obtains a set of imaginary proposal representations $\mathcal{F} = \{\mathbf{f}_i\}$ which allows Detection Head to learn object detection.

**Detection Head and Training Objectives**    We add an object detection head at the end of the Representation Generator, which takes the proposals $\mathcal{P}$ and corresponding representations $\mathcal{F}$ as input and predicts the bounding box of an object. IMAGINARYNET uses different Detection Head based on different task settings.

In ISOD, because no real images and annotations can be accessed, we learn the detection head following the OICR (Tang et al., 2017) method to predict the classification probability of each proposal, *i.e.*, $\hat{\mathbf{Y}} = f\left(\mathbf{I}_w^G; \theta^{\text{ISOD}}\right)$, only using the dataset $\mathcal{D}_w^G = \{(\mathbf{I}_w^G, \mathbf{Y}_w^G)\}$ generated by IMAGINARYNET. Here, $\mathbf{I}_w^G$ denotes an image generated by synethsis model, $\mathbf{Y}_w^G = \left[y_1^G, \ldots, y_{|\mathcal{C}|}^G\right]$ indicates the image-level weak-annotation. Following OICR, the object detection head consists of a Multiple Instances Learning (MIL) branch and a Refinement branch with the structure shown in Fig (2). During the

---

[1]We set $M^L$ as 15. We tried different $M^L$, but no significant gain was obtained.

Table 1: **Comparison of our IMAGINARYNET(ISOD) with baseline methods on PASCAL VOC 2007.** We consider two baseline methods, a strong baseline based on unsupervised CLIP model, and an upper-bound baseline based on OICR trained on 5,000 real images with image-level annotations.

| METHODS | AERO | BIKE | BIRD | BOAT | BOTTLE | BUS | CAR | CAT | CHAIR | COW | TABLE | DOG | HORSE | MOTOR | PERSON | PLANT | SHEEP | SOFA | TRAIN | TV | MAP |
|---|---|---|---|---|---|---|---|---|---|---|---|---|---|---|---|---|---|---|---|---|---|
| OICR (WSOD) | 54.97 | 55.32 | 47.64 | 29.25 | 24.94 | 69.32 | 64.76 | 76.07 | 18.16 | 56.59 | 20.17 | 70.13 | 69.03 | 64.42 | 19.82 | 20.12 | 49.14 | 27.42 | 68.61 | 52.72 | 47.93 |
| CLIP | 16.30 | 17.15 | 19.42 | 8.80 | 14.53 | 30.28 | 25.83 | 25.92 | **20.34** | 27.10 | **21.85** | 26.00 | 20.32 | 22.59 | 10.09 | 13.43 | 26.24 | 27.82 | 19.26 | **23.59** | 20.84 |
| IMAGINARYNET | **48.27** | **34.42** | **42.84** | **16.14** | **18.93** | **45.77** | **44.57** | **60.79** | 12.60 | **39.79** | 20.33 | **52.05** | **50.38** | **50.53** | **22.45** | **15.95** | **38.66** | **41.94** | **31.93** | 20.25 | **35.43** |

training phase, all object representations are first fed into the MIL branch. The MIL branch combines two parallel classification streams, each of which contains a softmax function. One softmax function is along the classes dimension to output the score matrix $\mathbf{x}_c^{\mathrm{m}} \in \mathbb{R}^{|\mathcal{P}| \times |\mathcal{C}|}$ , and the other is along the object dimension to output the score matrix $\mathbf{x}_r^{\mathrm{m}} \in \mathbb{R}^{|\mathcal{P}| \times |\mathcal{C}|}$. Subsequently, we compute the element-wise product of these two score matrixes and get the $\mathbf{x}^{\mathrm{m}} = \mathbf{x}_c^{\mathrm{m}} \odot \mathbf{x}_r^{\mathrm{m}}$. $\mathbf{x}^{\mathrm{m}}$ indicates the final object-level classification scores. Then we get the $c$-th class image-level score $s_c^{\mathrm{m}}$ by sum over all object: $s_c^{\mathrm{m}} = \sum_{i=1}^{|\mathcal{P}|} x_{i,c}^{\mathrm{m}}$. We convert the target label $c$ into one-hot form $\hat{\mathbf{y}}$ and calculate the Binary Cross Entropy (BCE) loss:

$$\mathcal{L}_{\mathrm{mil}}^{\mathrm{ign}} = -\sum_{c \in \mathcal{C}} \hat{y}_c \log s_c^{\mathrm{m}} + (1 - \hat{y}_c) \log (1 - s_c^{\mathrm{m}}). \tag{2}$$

To further improve the performance of the detector, we integrate the self-training methodology and construct the Refinement branch, which has a similar structure as the RoI head of the R-CNN-like detector (Ren et al., 2015). Following Tang et al. (2017), we keep the proposal with the highest classification score from $\mathbf{x}^{\mathrm{m}}$ as the pseudo-ground-truth (PGT), which are used to optimize the Refinement branch by a refinement loss $\mathcal{L}_{\mathrm{ref}}$. Finally, the learning objective of our object detection head of ISOD is:

$$\mathcal{L} = \mathcal{L}_{\mathrm{mil}} + \mathcal{L}_{\mathrm{ref}}. \tag{3}$$

The whole training pipeline of ISOD is shown in Fig. 2(a). Here we take the OICR as the example and it is feasible to employ other WSOD methods for the end of ISOD.

In WSOD, by gradually introducing real images and manual annotations, IMAGINARYNET can collaborate with other more supervision setting to boost detection performance. We learn the detection head by $\hat{\mathbf{Y}} = f(\mathbf{I}; \theta^{\mathrm{WSOD}})$ using generated dataset $\mathcal{D}_{\mathrm{w}}^{\mathrm{G}} = \{(\mathbf{I}_{\mathrm{w}}^{\mathrm{G}}, \mathbf{Y}_{\mathrm{w}}^{\mathrm{G}})\}$ and real dataset $\mathcal{D}_{\mathrm{w}}^{\mathrm{R}} = \{(\mathbf{I}_{\mathrm{w}}^{\mathrm{R}}, \mathbf{Y}_{\mathrm{w}}^{\mathrm{R}})\}$. During training, both real images and synthesized images are fed into the Representation Generator and object detection head in order and optimize these two modules via back-propagation. And we only need to slightly modify the training pipeline of object detection head. When real images and image-level annotations are available, as shown in Fig. 2(b) we can uniformly sample synthesized and real images and feed them into the WSOD training process.

In SSOD, the real labeled dataset $\mathcal{D}_{\mathrm{f}}^{R} = \{(\mathbf{I}_{\mathrm{f}}^{\mathrm{R}}, \mathbf{Y}_{\mathrm{f}}^{\mathrm{R}})\}$ and real unlabeled dataset $\mathcal{D}_{\mathrm{u}}^{\mathrm{R}} = \{(\mathbf{I}_{\mathrm{u}}^{\mathrm{R}})\}$ can be accessed. IMAGINARYNET further generates a dataset $\mathcal{D}_{\mathrm{w}}^{\mathrm{G}} = \{(\mathbf{I}_{\mathrm{w}}^{\mathrm{G}}, \mathbf{Y}_{\mathrm{w}}^{\mathrm{G}})\}$. For simplicity, we ignore $\mathbf{Y}_{\mathrm{w}}^{\mathrm{G}}$ and combine $\{(\mathbf{I}_{\mathrm{u}}^{\mathrm{R}})\}$ and $\{(\mathbf{I}_{\mathrm{w}}^{\mathrm{G}})\}$ to form the unlabeld dataset $\mathcal{D}_{\mathrm{u}}$. We learn the detection head by $\hat{\mathbf{Y}} = f(\mathbf{I}; \theta^{\mathrm{SSOD}})$ using the combination of unlabeled dataset $\mathcal{D}_{\mathrm{u}}$ and real labeled dataset $\mathcal{D}_{\mathrm{f}}^{\mathrm{R}} = \{(\mathbf{I}_{\mathrm{f}}^{\mathrm{R}}, \mathbf{Y}_{\mathrm{f}}^{\mathrm{R}})\}$. When both real images and box-level annotations are available, the detection head consists of a Teacher branch and a Student branch following Unbiased-teacher (Liu et al., 2021). As shown in Fig. 2(c), we simply leverage the teacher-student framework widely adopted in SSOD to train the detection head. The teacher detection network generates pseudo-ground-truth (PGT) for unlabeled real images and imaginary images to train the student detection network. In addition, the student is also trained on real images with box-level ground-truth (GT) by a supervised-learning manner (Ren et al., 2015).

### 3.4 INFERENCE PIPELINE

During inference, the language model, synthesis model and CLIP Filter will be abandoned. Taking a real image as the input, the Representation Generator generates proposal representations, and the Detection Head decodes them to predict the detection bounding box.

Table 2: **Comparison of IMAGINARYNET(WSOD) on PASCAL VOC 2007.** All models use 5,000 real data from VOC2007 and IMAGINARYNET(WSOD) leverages extra 5,000 imaginary samples. In comparison, IMAGINARYNET(WSOD) achieves state-of-the-art performance.

| METHODS | AERO | BIKE | BIRD | BOAT | BOTTLE | BUS | CAR | CAT | CHAIR | COW | TABLE | DOG | HORSE | MOTOR | PERSON | PLANT | SHEEP | SOFA | TRAIN | TV | mAP |
|---|---|---|---|---|---|---|---|---|---|---|---|---|---|---|---|---|---|---|---|---|---|
| WSDDN | 39.41 | 50.09 | 31.48 | 16.30 | 12.61 | 64.45 | 42.82 | 42.63 | 10.06 | 35.72 | 24.92 | 38.24 | 34.41 | 55.60 | 9.39 | 14.72 | 30.22 | 40.68 | 54.70 | 46.94 | 34.77 |
| OICR | 54.97 | 55.32 | 47.64 | 29.25 | 24.94 | 69.32 | 64.76 | 76.07 | 18.16 | 56.59 | 20.17 | 70.13 | 69.03 | 64.42 | 19.82 | 20.12 | 49.14 | 27.42 | 68.61 | 52.72 | 47.93 |
| W2N | 74.00 | 73.82 | 59.40 | 28.34 | 43.43 | **80.03** | 72.61 | 81.23 | 14.00 | **76.75** | 25.98 | 58.64 | 63.76 | **75.69** | 10.86 | 29.56 | **60.38** | **63.76** | 79.56 | 67.29 | 56.95 |
| IMAGINARYNET + WSDDN | 43.19 | 47.41 | 36.28 | 19.07 | 23.45 | 62.14 | 50.96 | 43.98 | 17.27 | 40.96 | 34.83 | 54.06 | 44.60 | 54.12 | 19.71 | 18.25 | 37.42 | 42.78 | 61.65 | 45.96 | 39.90 (+5.13) |
| IMAGINARYNET + OICR | 58.51 | 63.29 | 50.87 | 20.25 | 27.33 | 64.91 | 62.35 | 69.86 | 25.69 | 56.49 | **52.44** | 63.01 | 59.32 | 64.00 | 37.08 | 25.65 | 48.17 | 50.19 | 66.64 | 61.71 | 51.39 (+3.47) |
| IMAGINARYNET + W2N | **74.30** | **81.51** | **66.56** | **34.92** | **57.09** | 78.78 | **79.53** | **85.50** | **32.11** | 73.54 | 46.96 | **81.42** | **75.09** | 75.10 | **42.82** | **37.48** | 59.23 | 62.82 | **82.22** | **73.97** | **65.05 (+8.10)** |

# 4 EXPERIMENTS

## 4.1 IMPLEMENTATION DETAILS

We use GPT-2 (Radford et al., 2019) as the language model and DALLE-mini, which can better follow the language guidance, as the text-to-image synthesis model. We implement the image encoder with ResNet50 pretrained on ImageNet dataset. For Proposal Generator, we use Selective Search following W2N (Huang et al., 2022) for obtaining no real images and manual annotations. All training hyper-parameters follow OICR (Tang et al., 2017) and W2N (Huang et al., 2022) for a fair comparison. Unless otherwise specified, we generate 5,000 imaginary images during training, which has the similar amount of images in comparison to PASCAL VOC2007 trainval set.

## 4.2 EXPERIMENTAL SETUP

We first compare IMAGINARYNET with the ISOD model to verify whether it is feasible to learn object detectors without real images and manual annotations. To the best of our knowledge, we are among the first to investigate ISOD solely based on synthesized images. So we set up a strong baseline based on the CLIP model. For CLIP baseline, we use the Edge Boxes (Zitnick & Dollár, 2014) algorithm [2] to extract potential proposals on VOC test images for CLIP to classify its class, based on its similarity score with pre-designed prompts. Before computing AP, we apply the Non-Maximum Suppression (NMS) operation to remove redundant proposals based on CLIP scores. See Appendix A.1.1 for more details.

We also assess IMAGINARYNET by collaborating with WSOD method W2N (Huang et al., 2022), *i.e.*, IMAGINARYNET(WSOD). For comparison, we consider WSDDN (Bilen & Vedaldi, 2016), OICR (Tang et al., 2017) and W2N (Huang et al., 2022) on real images as the baseline detection networks. Compared with them, IMAGINARYNET in ISOD, *i.e.*, IMAGINARYNET(ISOD), adopts similar training pipeline but does not require real images and manual annotations, whereas IMAGINARYNET(WSOD) leverages both real and synthesized images.

## 4.3 OVERALL RESULTS

### 4.3.1 IMAGINARY-SUPERVISED OBJECT DETECTION

As shown in Tab. 1, IMAGINARYNET(ISOD) achieves an mAP of 35.43. This verifies the feasibility of learning object detectors without real images and annotations. Moreover, IMAGINARYNET(ISOD) outperforms the CLIP baseline model with a gain of 14.59 in mAP. This further supports that the competitiveness of a pre-trained generative model against pre-trained contrastive alignment models (*e.g.*, CLIP) for learning object detectors. For most classes, IMAGINARYNET(ISOD) can obtain a significant performance gain, and IMAGINARYNET(ISOD) has much fewer parameter amounts and computational cost in inference because the pre-trained modules can be abandoned after training. Especially, for some classes such as cat, the performance improved from 25.92 of CLIP to 60.79 of IMAGINARYNET(ISOD). The results clearly show the feasibility of IMAGINARYNET(ISOD), *i.e.*, learning object detectors without real images and manual annotations.

We further compare IMAGINARYNET(ISOD) with an upper-bound baseline, WSOD on real images with class-level annotations. For a fair comparison, we only generate 5,000 imaginary samples during training. Compared with OICR, which has the same backbone with ours and trained on 5,000 real images and manual annotations in WSOD setting, we achieved about 75% performance of it.

---

[2]We tried Selective Search but Edge Boxes is more effective, so we report the best results.

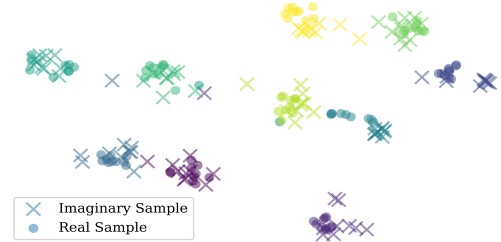

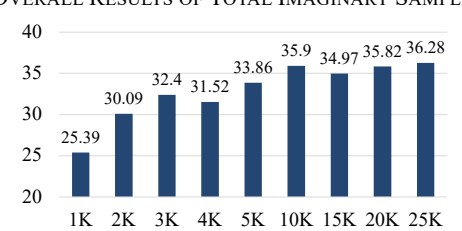

OVERALL RESULTS OF TOTAL IMAGINARY SAMPLES

Figure 3: **Visualization of extracted features.** We can observe that features can cluster based on the object type. This shows imaginary data contains similar knowledge like real data.

Figure 4: **Overall results of different total imaginary samples.** The performance improves steadily with the growth of imaginary samples.

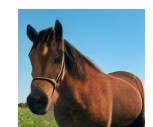
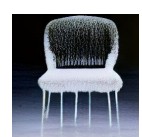
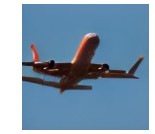
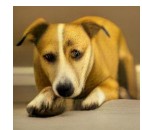

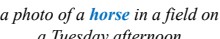

*a photo of a **horse** in a field on a Tuesday afternoon*  *a photo of a **chair**, which was covered by a thin layer of ice*  *a photo of a **aeroplane**, like a big big red plane*  *a photo of a **dog** that's in pain*

Figure 5: **Visualization of imaginary images.** Language model extends the prefix to a full description of the scene and the synthesis model generated images followed the description.

Table 3: **Comparison of our method with real data on PASCAL VOC 2007.** Imaginary data can play a similar role in enhancing performance. 10K Imaginary data can improve the model by a similar margin compared with 10K real data. Moreover, introducing imaginary data when extra real data exists can further improve the performance.

| METHODS | ANNOTATED REAL DATA | UN-ANNOTATED REAL DATA | UN-REAL DATA | mAP |
|---|---|---|---|---|
| FASTERRCNN | 5K VOC2007 | ∅ | ∅ | 76.00 |
| UNBIASED TEACHER | 5K VOC2007 | 10K VOC2012 | ∅ | 81.08 (+5.08) |
| IMAGINARYNET(SSOD) | 5K VOC2007 | ∅ | 5K IMAGINARY | 80.36 (+4.36) |
| IMAGINARYNET(SSOD) | 5K VOC2007 | ∅ | 10K IMAGINARY | 80.60 (+4.60) |
| IMAGINARYNET(SSOD) | 5K VOC2007 | 10K VOC2012 | 10K IMAGINARY | 81.60 (+5.60) |

Considering that synthesized images and image-level annotations are free for IMAGINARYNET, our method has the potential to further improve detection accuracy by increasing training samples.

**MSCOCO Results**  We also conduct the experiments on MSCOCO dataset, and obtain similar observations as VOC2007. See Appendix A.2.1 for more details.

### 4.3.2 INCORPORATING WITH WSOD ON REAL IMAGES AND MANUAL ANNOTATIONS

We further show that the synthesized images by IMAGINARYNET can collaborate with other supervision settings on real images and manual annotations for boosting detection performance. Here we use WSOD as an example. As given in Tab. 2, IMAGINARYNET(WSOD) has achieved state-of-the-art performance in WSOD. This shows the effectiveness of IMAGINARYNET can not only learn object detectors without real images and annotations, but also is effective in improving the performance of backbones without increasing parameters and inference cost. IMAGINARYNET(WSOD) obtains an mAP of 65.05, which is higher than the state-of-the-art WSOD models.

### 4.4 COMPARISON WITH REAL DATA

How effective is IMAGINARYNET in comparison with real data? To answer this question, we conduct comparison experiments in SSOD setting. We use the same backbone, RESNET50-FPN as the supervised model FASTERRCNN as well as the SSOD model UNBIASED TEACHER[3]. For fairness, we use the same backbone, hyper-parameters, and real data by following the competing methods.

---

[3]Although the performance in its original paper is lower, we obtain 81.08 by re-training the model.

Table 4: **Comparison of different imaginary samples in ISOD on PASCAL VOC 2007.** We set the different scales of samples and report their mAP. The mAP increases with the growth of samples.

| SAMPLES | AERO | BIKE | BIRD | BOAT | BOTTLE | BUS | CAR | CAT | CHAIR | COW | TABLE | DOG | HORSE | MOTOR | PERSON | PLANT | SHEEP | SOFA | TRAIN | TV | mAP |
|---|---|---|---|---|---|---|---|---|---|---|---|---|---|---|---|---|---|---|---|---|---|
| 1K | 40.26 | 32.42 | 29.63 | 4.00 | 10.83 | 26.56 | 35.23 | 35.67 | 10.28 | 31.48 | 8.78 | 37.78 | 32.23 | 43.38 | 18.60 | 13.89 | 27.81 | 36.72 | 12.97 | 19.22 | 25.39 |
| 2K | 42.79 | 39.76 | 38.41 | 4.22 | 15.97 | 41.26 | 40.11 | 49.16 | 11.72 | 37.18 | 5.91 | 48.31 | 41.80 | 44.57 | 19.23 | 14.15 | 33.57 | 38.11 | 19.34 | 16.14 | 30.09 |
| 3K | 47.18 | 39.48 | 38.64 | 6.11 | 17.18 | 37.39 | 39.47 | 62.44 | 11.37 | 37.67 | 11.21 | 47.27 | 52.55 | 44.82 | 20.49 | 14.51 | 39.29 | 37.80 | 19.04 | 24.00 | 32.40 |
| 4K | 44.19 | 39.48 | 43.04 | 5.80 | 18.44 | 33.64 | 40.24 | 43.19 | 11.68 | 36.69 | 13.23 | 39.56 | 54.83 | 51.47 | 19.00 | 15.77 | 36.75 | 42.77 | 21.65 | 19.04 | 31.52 |
| 5K | 48.69 | 39.83 | 43.95 | 6.11 | 16.92 | 34.51 | 42.34 | 60.56 | 11.66 | 40.87 | 17.38 | 41.24 | 49.83 | 52.74 | 19.89 | 16.00 | 38.01 | 43.93 | 23.24 | **29.50** | 33.86 |
| 10K | 47.79 | 41.32 | 43.82 | 10.09 | 18.74 | **42.96** | 41.02 | **64.34** | 12.62 | **45.38** | 24.89 | 49.47 | 50.29 | **53.61** | 20.90 | 15.02 | 40.10 | 43.54 | 24.25 | 27.86 | 35.90 |
| 15K | **49.25** | 38.94 | 44.23 | 16.86 | 23.95 | 36.01 | 41.72 | 45.01 | 13.62 | 42.15 | 21.54 | **53.59** | 48.23 | 49.97 | 22.56 | **18.35** | **48.45** | 46.68 | 27.85 | 10.42 | 34.97 |
| 20K | 48.44 | 41.09 | 43.20 | 16.29 | **25.04** | 40.14 | **43.49** | 43.83 | 13.97 | 40.10 | **25.96** | 49.82 | 51.14 | 51.85 | **22.68** | 17.46 | 44.08 | **48.45** | **31.02** | 18.28 | 35.82 |
| 25K | 49.11 | **41.34** | **46.78** | **17.02** | 21.09 | 33.98 | 41.42 | 61.53 | **14.43** | 43.63 | 22.27 | 51.88 | **55.27** | 51.45 | 21.54 | 16.47 | 46.63 | 46.51 | 26.06 | 17.21 | **36.28** |

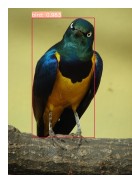 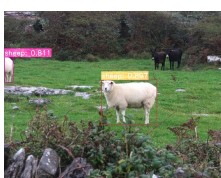 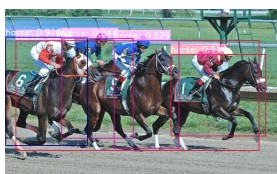 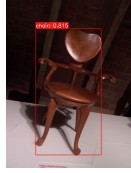 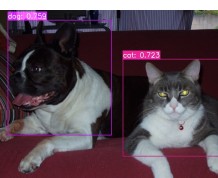

Figure 6: **Visualization of detection results on PASCAL VOC 2007 test split.** Results show that it is feasible to learn object detectors via IMAGINARYNET(ISOD).

As shown in Tab. 3, comparing with FASTER-RCNN and UNBIASED TEACHER, IMAGINARYNET(SSOD) has achieved comparable performance, about 99.4% performance of UNBIASED TEACHER. This shows that generated data are almost as effective as real data but have much lower cost in data acquisition and higher controllability in class balance. Compared to the baseline UNBIASED TEACHER, we can observe that generated data are orthogonal with real data, where we can improve the performance by increasing both real and synthesized data.

## 4.5 VISUALIZATION

### 4.5.1 EXTRACTED FEATURES

To verify whether the features extracted from imaginary images contain similar information to real images, we extracted their features from the same Representation Generator. For real images, we extract 10 features per class as anchors and 10 features per class from imaginary images. We use T-SNE to project them to a 2D image.

In Fig. 3, we can observe that features of one class from imaginary images are coincident with the feature cluster of the same class from real images. This indicates that features from imaginary images contain similar information in comparison with real images, which explains why IMAGINARYNET can learn object detectors even without real images and annotations.

### 4.5.2 IMAGINARY IMAGES

We randomly sample imaginary images from the synthesis model and corresponding text descriptions from the language model to observe how they can play the role of real data. In Fig. 5, we can find that text descriptions are quite diverse. Meanwhile, high-quality images from the text-to-image synthesis model follow the description well and contain vivid objects of the corresponding class. Although all images generated are not in real datasets, IMAGINARYNET successfully simulates the scene of the images from object detection datasets. This may explain why IMAGINARYNET can even enhance object detectors under settings that contain real images and annotations.

### 4.5.3 CASE STUDIES

To better reveal the quality of IMAGINARYNET, we visualize detection results from VOC2007 test split in ISOD setting. In Fig. 6, many objects are detected correctly. In some hard cases, *e.g.*, the 5-th image which contains many drivers and motors overlapping together, can also be detected well with nearly-correct boxes and classes. This proves the effectiveness of IMAGINARYNET in ISOD.

Although IMAGINARYNET shows pretty effectiveness in many cases, we still find out some failure cases. For example, in Fig. 6, some motorbikes are not detected well and persons are classified into the wrong class. This is because ISOD is a very strict setting, where the model cannot access any

Table 5: **Ablation of components in ISOD on PASCAL VOC 2007.** We can observe that the language model improve the results of the model significantly.

| METHODS | AERO | BIKE | BIRD | BOAT | BOTTLE | BUS | CAR | CAT | CHAIR | COW | TABLE | DOG | HORSE | MOTOR | PERSON | PLANT | SHEEP | SOFA | TRAIN | TV | MAP |
|---|---|---|---|---|---|---|---|---|---|---|---|---|---|---|---|---|---|---|---|---|---|
| IMAGAINARYNET | **48.27** | 34.42 | **42.84** | **16.14** | 18.93 | **45.77** | **44.57** | 60.79 | 12.60 | 39.79 | **20.33** | **52.05** | **50.38** | 50.53 | **22.45** | 15.95 | 38.66 | 41.94 | **31.93** | 20.25 | **35.43** |
| W/O CLIP FILTER | 47.18 | 43.13 | 35.51 | 5.33 | 19.23 | 32.58 | 41.23 | **66.72** | 12.03 | **43.98** | 18.13 | 43.98 | 36.71 | **51.68** | 19.04 | 14.97 | 34.12 | **42.74** | 23.04 | 33.36 | 33.23 |
| W/O LANGUAGE MODEL | 31.32 | **49.94** | 32.13 | 3.24 | **23.78** | 43.35 | 34.66 | 30.95 | **13.35** | 38.59 | 15.77 | 38.51 | 40.82 | 42.34 | 15.84 | **18.39** | **42.11** | 38.86 | 18.99 | **47.54** | 31.02 |

real images and annotations. Nonetheless, IMAGINARYNET still shows that it is feasible to learn object detectors in this hard setting. We further analyze this in Appendix A.3.

## 5 ABLATION STUDIES

### 5.1 INFLUENCE OF TOTAL IMAGINARY SAMPLES

Can the model obtain higher performance if IMAGINARYNET imagine more samples? To figure out this, we conduct experiments on ISOD with different scales of imaginary samples. In Tab. 4 and Fig. 4, one can see that the overall performance, mAP, increases steadily to 36.28 finally. The result shows that more imaginary samples can make the detector stronger. Meanwhile, IMAGINARYNET obtains mAP of 25.39 with 1K samples only. This shows the effectiveness of imaginary samples, which can train the detector with very limited samples. We also notice the fluctuation of several classes, which may be explained by the instability of class-level annotation learning. In general, more samples will improve the final performance.

### 5.2 EFFECTIVENESS OF COMPONENTS

Whether the various scene descriptions and error samples affect detector learning? To figure out this question, we re-train an IMAGINARYNET without the language model and CLIP Filter. In Tab. 5, we can observe a significant drop in mAP when no language model and CLIP Filter existing. Without CLIP filter, the average performance drops significantly. This show the that error samples may affect the final results. Without language model, the performance of each class also drops in most cases. Such class, like cat, drops over 30 of AP50. The result indicates that language model is also important in generating diverse images. We also notice the unexpected drop in some classes and we believe this can be explained by the failure of the language model to generate a scene similar to the real dataset in some cases or that CLIP reduce the diversity of generated data. We will leave this as future work to generate images that are more suitable for learning detectors.

## 6 BROADER IMPACT

IMAGINARYNET is an effective framework to learn detectors without real images and annotations for further improving the performance with real data. It can provide new insight for handling complex vision tasks under low-resource regime. For example, it is possible to learn detection or segmentation models without any real data. However, the performance of IMAGINARYNET(ISOD) is still inferior to WSOD and FSOD. Thus, more studies are required to generate diverse and photo-realistic images and to better extract supervision information from the language descriptions and synthesized images. We discuss the limitations in Appendix A.5.

## 7 CONCLUSION

In this paper, we proposed a novel learning paradigm of object detection, Imaginary-Supervised Object Detection (ISOD), where no real images and manual annotations can be used for training object detectors. In particular, we presented an IMAGINARYNET framework to generate synthesized images as well as supervision information. Experiments shows that IMAGINARYNET(ISOD) can obtain about 75% of performance in comparison with the WSOD counterpart trained using real data and image-level annotations. Moreover, IMAGINARYNET can collaborate with other supervision settings on real data to further boost detection performance.

ACKNOWLEDGEMENT

This work was supported by the National Key RD Program of China under Grant No. 2021ZD0112100.

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

## A APPENDIX

### A.1 ADDITIONAL IMPLEMENTATION DETAILS

#### A.1.1 CLIP BASELINE

We use ViT-B/16 as the image encoder for CLIP and use Edge Boxes to extract 1000 proposals for each test image. Each proposal will be transformed to $384 \times 384$ to fit the CLIP input size. Then each proposal will be calculated similarity with the prompt which consists of the prefix "a photo of" and the class name. We select the highest CLIP similarity as the class of the proposal. Before computing AP, we apply the Non-Maximum Suppression (NMS) operation to remove redundant proposals according to their CLIP scores.

#### A.1.2 CLASS SELECTING IN WSOD

To obtain a better performance in WSOD, the model trained with selected classes of Imaginary samples instead of all Imaginary samples. We first trained the model with all imaginary samples and evaluate the performance of each class in val split. We selected classes that the performance is higher than baseline model. Then we re-trained the model with imaginary samples of selected classes to obtain the best performance. As the results shown in Tab. 6, we can observe that the IMAGINARYNET obtains a better performance even without class selecting. Combining with class selecting, we obtain the best performance. We explained this as the gap among Imaginary samples and real samples in some classes, such as boat or train.

Table 6: **Comparison of data used in WSOD on PASCAL VOC 2007.** We can observe that generating selected class improve the performance of IMAGINARYNET significantly.

| METHODS | CLASS SELECTING | AERO | BIKE | BIRD | BOAT | BOTTLE | BUS | CAR | CAT | CHAIR | COW | TABLE | DOG | HORSE | MOTOR | PERSON | PLANT | SHEEP | SOFA | TRAIN | TV | mAP |
|---|---|---|---|---|---|---|---|---|---|---|---|---|---|---|---|---|---|---|---|---|---|---|
| OCIR | | 54.97 | 55.32 | 47.64 | **29.25** | 24.94 | **69.32** | **64.76** | **76.07** | 18.16 | **56.59** | 20.17 | **70.13** | **69.03** | 64.42 | 19.82 | 20.12 | 49.14 | 27.42 | **68.61** | 52.72 | 47.93 |
| IMAGINARYNET + OCIR | | 56.32 | 61.65 | 50.59 | 24.12 | 25.28 | 64.65 | 62.61 | 69.84 | **26.04** | 55.95 | 45.49 | 53.34 | 63.27 | 63.46 | **38.75** | 23.87 | **50.68** | **50.75** | 43.49 | 59.76 | 49.49 |
| IMAGINARYNET + OCIR | ✓ | **58.51** | **63.29** | **50.87** | 20.25 | **27.33** | 64.91 | 62.35 | 69.86 | 25.69 | 56.49 | **52.44** | 63.01 | 59.32 | 64.00 | 37.08 | **25.65** | 48.17 | 50.19 | 66.64 | **61.71** | **51.39** |

### A.2 EXTRA COMPARISONS

#### A.2.1 COMPARISON IN IMAGINARY-SUPERVISED OBJECT DETECTION ON MSCOCO

Table 7: **Comparison of our method in ISOD on MSCOCO2014.** We obtain a much higher mAP compared with the CLIP baseline. This shows the feasibility of training the detector on a harder dataset of 80 classes without any real images and annotations.

| METHODS | mAP |
|---|---|
| CLIP | 6.3 |
| IMAGINARYNET | **11.4** |

We also verified our framework on MSCOCO2014, which is also a common dataset in object detection. As the results in Tab. 7, we outperform the CLIP baseline with nearly two times of mAP. This proves that our framework is also effective on the harder dataset, which contains 80 classes.

#### A.2.2 COMPARISON WITH FINE-TUNED CLIP

Table 8: **Comparison with fine-tuned CLIP.** IMAGINARYNET still outperforms CLIP even after fine-tuning CLIP on our generated data.

| METHODS | AERO | BIKE | BIRD | BOAT | BOTTLE | BUS | CAR | CAT | CHAIR | COW | TABLE | DOG | HORSE | MOTOR | PERSON | PLANT | SHEEP | SOFA | TRAIN | TV | mAP |
|---|---|---|---|---|---|---|---|---|---|---|---|---|---|---|---|---|---|---|---|---|---|
| CLIP | 16.30 | 17.15 | 19.42 | 8.80 | 14.53 | 30.28 | 25.83 | 25.92 | **20.34** | 27.10 | **21.85** | 26.00 | 20.32 | 22.59 | 10.09 | 13.43 | 26.24 | 27.82 | 19.26 | 23.59 | 20.84 |
| CLIP + FINETUNE | 18.74 | 20.37 | 15.72 | 5.10 | 9.78 | 32.86 | 35.50 | 27.98 | 12.19 | 23.79 | 12.69 | 41.42 | 29.17 | 14.91 | **24.07** | 6.29 | 34.89 | 29.02 | 21.66 | **24.04** | 22.01 |
| IMAGINARYNET | **48.27** | **34.42** | **42.84** | **16.14** | **18.93** | **45.77** | **44.57** | **60.79** | 12.60 | **39.79** | 20.33 | **52.05** | **50.38** | **50.53** | 22.45 | **15.95** | **38.66** | **41.94** | **31.93** | 20.25 | **35.43** |

We think that CLIP cannot use the information from the overlapping proposals because its pre-training is based on image-text matching. But the detection head of IMAGINARYNET is designed for the training detection task. We conduct the extra comparison by fine-tuning the CLIP model on the same data we used for IMAGINARYNET.

As the Tab. 8, IMAGINARYNET still outperforms CLIP even after fine-tuning CLIP on our generated data. But CLIP is still a strong baseline for this task because it uses a large-scale pre-training and can be used to make a comparison in ISOD, the strict setting that models cannot access any real images and annotations.

### A.2.3 COMPARISON WITH OTHER CLIP-BASED DETECTION APPROACHES

We add the extra comparison with these CLIP-based Detection approaches. Due to these models, e.g., RegionCLIP (Zhong et al., 2022b), Detic (Zhou et al., 2022), and ViLD (Gu et al., 2021), use different human-annotated real detection data, we adopt the labeled dataset for training each model. For a fair comparison, we adopt IMAGINARYNET(SSOD) for leveraging both imaginary data and the same labeled data. We compare results on 20 base class in VOC 2007 test split. For ViLD, we directly provide the results from their paper.

Table 9: **Comparison with other clip-based object detection approaches on PASCAL VOC 2007.** IMAGINARYNET still obtains the best performance.

| METHODS | ANNOTATED REAL DATA | UNANNOTATED REAL DATA | UNANNOTATED IMAGINARY DATA | MAP |
|---|---|---|---|---|
| REGIONCLIP | 74K COCO-48 | 1.5M CC | ∅ | 68.35 |
| DETIC | 165K COCO + 100K LVIS | 14M IMAGENET | ∅ | 79.44 |
| VILD | 5K VOC 2007 + 100K LVIS | ∅ | ∅ | 78.9 |
| IMAGINARYNET(SSOD) | 5K VOC 2007 | ∅ | 5K IMAGINARY | 80.36 |
| IMAGINARYNET(SSOD) | 5K VOC 2007 | ∅ | 10K IMAGINARY | 80.60 |
| IMAGINARYNET(SSOD) | 5K VOC 2007 | 10K VOC 2012 | 10K IMAGINARY | 81.60 |

From the Tab. 9, we can observe IMAGINARYNET(SSOD) still obtains the best performance. This proves the effectiveness of generated data from IMAGINARYNET.

### A.3 FAILURE CASE

From Fig. 8, we can observe that WSOD can better detect objects that are usually accompanying with each other, e.g., rider and horse. This is because we cannot know all the categories appearing in the generation image precisely. For example, we cannot know whether the generated images contain only a motorbike or a human riding a motorbike. If two objects are not accompanying usually, this may not influence the final results because of plenty of training samples. However, for objects accompanying usually, it is easy for the model to consider them as a whole object. This may motivate us to explore how to control the accompanying in the generation or better learn accompanying objects. A modified cross-attentive module for focusing on the key object described in the text may help the model to decrease the possibilities of generating accompanying objects

### A.4 DISCUSSION

Table 10: **Comparison of different prefix on PASVAL VOC 2007.** Different prefixes can affect the final performance of detection.

| METHODS | AERO | BIKE | BIRD | BOAT | BOTTLE | BUS | CAR | CAT | CHAIR | COW | TABLE | DOG | HORSE | MOTOR | PERSON | PLANT | SHEEP | SOFA | TRAIN | TV | MAP |
|---|---|---|---|---|---|---|---|---|---|---|---|---|---|---|---|---|---|---|---|---|---|
| PREFIX 1 | 47.18 | 43.13 | 35.51 | 5.33 | 19.23 | 32.58 | 41.23 | 66.72 | 12.03 | 43.98 | 18.13 | 43.98 | 36.71 | 51.68 | 19.04 | 14.97 | 34.12 | 42.74 | 23.04 | 33.36 | 33.23 |
| PREFIX 2 | 40.77 | 48.30 | 35.59 | 7.33 | 23.73 | 45.56 | 38.24 | 45.30 | 13.83 | 35.68 | 15.73 | 38.39 | 41.82 | 42.24 | 12.95 | 16.47 | 39.64 | 37.34 | 24.57 | 42.48 | 32.30 |
| PREFIX 3 | 13.22 | 47.62 | 29.16 | 6.24 | 21.62 | 42.10 | 35.94 | 26.42 | 11.99 | 34.50 | 14.37 | 34.87 | 36.68 | 44.39 | 14.14 | 19.04 | 38.32 | 35.41 | 22.38 | 40.42 | 28.44 |

### A.4.1 PREFIX

Prefix is the initial input, which can provide important cues and the language model will extend the prefix to a longer sentence. Prefix is used in many zero-shot language generation or vision-and-language works (Radford et al., 2021; Wang et al., 2021; Cho et al., 2021). The prefix will provide important cues for language models. Different prefixes will change the results of the language model and the results of the text-to-image synthesis model then as well. It will also affect the final performance of detection.

Many works (Radford et al., 2021; Ma et al., 2021; Xu et al., 2022) prove that some simple prefixes can help the multimodal model calculate vision-and-language information better. Following these

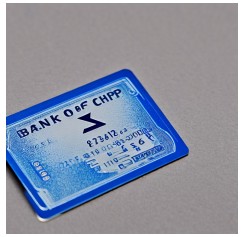 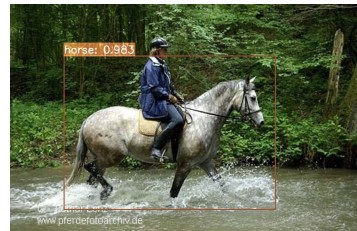 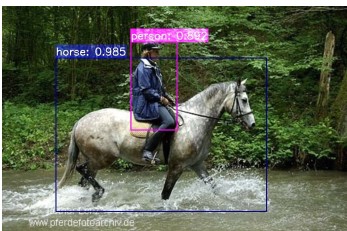

Figure 7: **Composition and extrapolation ability of text-to-image synthesis model.** The model can create an object with the description only.

Figure 8: **Failure case in ISOD.** Compared the result of IMAGI-NARYNET in ISOD (left) with the result of OICR in WSOD (right), we can observe that IMAGINARYNET detect the rider and horse as a whole object. Model of WSOD can better detect objects that are usually accompanying with each other. This may motivate us to explore how to control the accompanying in the generation or better learn accompanying objects.

previous works, we consider four representative prefixes as follow: (1) "A photo of a {class}", (2) "A photograph of a {class}", (3) "An image of a {class}", and (4) "A {class}".

We conduct the comparison on prefixes in Tab. 10. Each prefix will generate 5K samples for training. To avoid influence from CLIP, which may make the model re-generate images forever when prefix is bad, we disable CLIP Filter in this experiment.

For the Prefix (1) and (2), the language model can generate the sentence that describes the content of images based on the prefix. Meanwhile, the word in these prefixes, "photo" or "photograph", forces the text-to-image synthesis model to generate real-style images like photos. This is also the way for us to control the domain gap between the generated images and real images in the dataset.

Prefix (3) obtains a much worse result. We can observe that although the language model still can generate the sentence describing objects well, the text-to-image synthesis model generates many images of cartoon or art style. This leads to a domain gap that makes the model get worse results. Some classes that have a significant gap in real and art style, such as cat, drop by a huge margin.

Prefix (4) is an even worse prefix than (3). Most sentences generated by the language model are not describing the object, e.g., "A person should seek professional assistance to obtain an accommodation.". We did not train our model based on the prefix (4) because it is hard for the text-to-image synthesis model to generate an image based on such a sentence without any object describing.

### A.4.2 SYNTHESIS MODEL

Different from class-to-image synthesis models, text-to-image synthesis models have a much stronger capability of composition and extrapolation (Ramesh et al., 2021; Rombach et al., 2022; Saharia et al., 2022). For example, the model may be trained with only "black cats" and "white swans", but it can generate "black swans". Meanwhile, text-to-image synthesis models can even create new objects with detailed descriptions. For example, they can generate the image with the description "A photo of blue card on the table with 1mm thickness and rounded corner, card numbers in the card with title 'Bank of Card'" (see Fig. 7).

Table 11: **Comparison of different visual synthesis models in ISOD on PASCAL VOC 2007.** We can observe that DALLE-MINI can provide higher performance for IMAGINARYNET.

| METHODS | AERO | BIKE | BIRD | BOAT | BOTTLE | BUS | CAR | CAT | CHAIR | COW | TABLE | DOG | HORSE | MOTOR | PERSON | PLANT | SHEEP | SOFA | TRAIN | TV | MAP |
|---|---|---|---|---|---|---|---|---|---|---|---|---|---|---|---|---|---|---|---|---|---|
| STABLE DIFFUSION | 35.71 | 24.26 | 13.67 | 14.96 | 1.17 | **34.72** | 31.76 | 41.43 | 9.76 | 20.66 | 25.39 | 36.04 | 27.37 | 35.90 | 10.46 | 3.78 | 17.50 | 36.68 | 30.89 | 13.53 | 23.28 |
| DALLE-MINI | **47.18** | **43.13** | **35.51** | **5.33** | **19.23** | 32.58 | **41.23** | **66.72** | **12.03** | **43.98** | **18.13** | **43.98** | **36.71** | **51.68** | **19.04** | **14.97** | **34.12** | **42.74** | **23.04** | **33.36** | **33.23** |

We notice that the generated images are different from different generative models. In this paper, we use two effective open-source visual synthesis models, STABLE DIFFUSION and DALLE-MINI. We conduct a comparison on ISOD to verify the influence of synthesis models. In Tab. 11, we can find that DALLE-MINI can provide higher performance for IMAGINARYNET. We explain this as DALLE-MINI's results contain fewer objects, which are easier for the model to learn in the training.

Table 12: **Ablation of data augmentation strategies training on PASCAL VOC 2007.** We can observe that all augmentation strategies are effective on imaginary samples.

| Train Aug. | Test Aug. | Aero | Bike | Bird | Boat | Bottle | Bus | Car | Cat | Chair | Cow | Table | Dog | Horse | Motor | Person | Plant | Sheep | Sofa | Train | TV | mAP |
|---|---|---|---|---|---|---|---|---|---|---|---|---|---|---|---|---|---|---|---|---|---|---|
| | | 86.05 | 86.75 | 81.92 | 66.66 | 71.11 | 86.72 | 88.13 | 87.68 | 61.24 | 84.58 | 74.15 | 87.37 | 86.73 | 86.95 | 85.65 | 56.45 | 81.45 | 76.94 | 85.03 | 76.89 | 79.92 |
| √ | | 85.35 | 87.56 | 81.27 | 74.48 | 70.22 | 86.46 | 88.66 | 88.35 | 61.90 | 85.04 | 76.49 | 86.39 | 87.63 | 88.26 | 85.68 | 55.87 | 79.98 | 77.51 | 86.13 | 78.86 | 80.60 |
| √ | √ | 87.83 | 88.55 | 84.83 | 76.29 | 76.08 | 87.68 | 89.19 | 88.28 | 66.40 | 87.34 | 75.66 | 88.52 | 88.77 | 88.62 | 86.58 | 62.33 | 82.23 | 79.86 | 87.71 | 81.22 | 82.70 |

### A.4.3 ORTHOGONALITY WITH DATA AUGMENTATIONS

Table 13: **The performance of ImaginaryNet with Active Selection strategy on PASCAL VOC 2007 in ISOD.** We can observe a significant gain with this loop.

| Methods | Aero | Bike | Bird | Boat | Bottle | Bus | Car | Cat | Chair | Cow | Table | Dog | Horse | Motor | Person | Plant | Sheep | Sofa | Train | TV | mAP |
|---|---|---|---|---|---|---|---|---|---|---|---|---|---|---|---|---|---|---|---|---|---|
| ImaginaryNet | 49.25 | 38.94 | 44.23 | 16.86 | 23.95 | 36.01 | 41.72 | 45.01 | 13.62 | 42.15 | 21.54 | 53.59 | 48.23 | 49.97 | 22.56 | 18.35 | 48.45 | 46.68 | 27.85 | 10.42 | 34.97 |
| ImaginaryNet + Active Selection | 46.78 | 43.80 | 43.65 | 12.56 | 24.75 | 55.04 | 42.33 | 36.58 | 15.54 | 44.51 | 23.76 | 52.32 | 53.84 | 50.42 | 23.94 | 20.26 | 48.16 | 47.23 | 28.67 | 30.95 | 37.25 |

Whether the data augmentation strategies still work on imaginary images? To answer the question, we conduct several ablation studies with different data augmentation strategies, training augmentation, and testing augmentation. For most WSOD methods, which are also employed into our ISOD setting, no training augmentation is needed because it is relatively hard to learn with class-labeled images. Consequently, we conduct this experiment in SSOD setting. As Tab. 12 shows, we can observe that IMAGINARYNET(SSOD) obtains better performance after activating training augmentation and testing augmentation. These results give strong evidence that our framework is orthogonal with data augmentations. Moreover, orthogonality with data augmentation strategies may imply that imaginary samples have a similar property to real data.

### A.4.4 ACTIVE SELECTION

We setup an Active Selection strategy by using the detector to find out the samples generated which have the highest loss in a group of samples. We train IMAGINARYNET on 10K data first to initialize it and then continue to optimize it with the loop mentioned above on 5K extra samples. The generator will generate 15 samples, and the detector will choose 5K samples with the highest loss. To make a fair comparison, we use an IMAGINARYNET trained with 15K data as the baseline.

From the Tab. 13, we can observe a significant gain by taking into account this Active Sample Selecting strategy. We think the reason for the improvement is similar to the GAN-based network: the goal of the detector is to minimize the loss of the samples, but the goal of the generator is to generate the sample that makes the detector have the highest loss. These adversarial goals make the network more effective.

### A.5 LIMITATIONS

We further discuss the limitations and conduct extra ablation studies on the quantity and quality of the generated images from three aspects: (1) sample number, (2) diversity and photo-reality, and (3) generation errors.

### A.5.1 SAMPLE NUMBER

As the Tab. 4, one can see that IMAGINARYNET obtains the mAP of 36.28 when using 25K generated samples. Nonetheless, the growth of performance has slowed down when the number of generated samples is more than 10K. Thus, we speculate that 36.28 approaches the upper bound of the performance. We note that it may be hard for our method to surpass the method training on full annotated real images. How to utilize the generative models effectively for refreshing this upper bound will be a question in the future study.

### A.5.2 DIVERSITY AND PHOTO-REALITY

The diversity of images generated will help the model learn more knowledge and improve its robustness. Diversity is affected by not only the text-to-image synthesis model but also language prompts. This is also the motivation that we introduce the language model to the framework to obtain better diversity. Meanwhile, higher photo-reality also helps the model to obtain a better performance due to low domain gap to real images. With the incredible increase in text-to-image synthesis models,

images that are very similar to real photos can be synthesized. Therefore, we believe that we can obtain images generated with higher and higher photo-reality.

Table 14: **Quantification of language and text-to-image generation errors by Clip.**

| METHODS | AERO | BIKE | BIRD | BOAT | BOTTLE | BUS | CAR | CAT | CHAIR | COW | TABLE | DOG | HORSE | MOTOR | PERSON | PLANT | SHEEP | SOFA | TRAIN | TV | MAP |
|---|---|---|---|---|---|---|---|---|---|---|---|---|---|---|---|---|---|---|---|---|---|
| ERROR NUM | 35 | 82 | 147 | 87 | 112 | 310 | 115 | 95 | 219 | 163 | 87 | 160 | 117 | 116 | 451 | 53 | 114 | 140 | 155 | 689 | 3447 |
| ERROR RATIO | 2.80% | 6.56% | 11.76% | 6.96% | 8.96% | 24.80% | 9.20% | 7.60% | 17.52% | 13.04% | 6.96% | 12.80% | 9.36% | 9.28% | 36.08% | 4.24% | 9.12% | 11.20% | 12.40% | 55.12% | 13.79% |

### A.5.3 GENERATION ERRORS

As the Tab. 14, we find that we have a 13.79% error rate overall and each class has a different error rate. This limits the performance of IMAGINARYNET. By introducing CLIP Filter in IMAGI-NARYNET, we can increase its performance significantly but the generation errors may still be left in some samples.

