# OpenReview forum: "ImaginaryNet: Learning Object Detectors without Real Images and Annotations"
_ICLR.cc/2023/Conference — ICLR 2023 poster_

### Official Review · Reviewer_va3v · 2022-10-17

**Confidence:** 4
**Correctness:** 3
**Technical Novelty And Significance:** 3
**Empirical Novelty And Significance:** 2
**Recommendation:** 6

**Clarity, Quality, Novelty And Reproducibility:**

- The overall idea looks somewhat novel to me, although there are im2real based on Gaming engine before, but using image-to-text, to the best my knowledge, this is the first.

- Although the paper just consists of several public available modules such as GPT-2, DALLE-MINI, WSOD, it still lacks of details or detailed studies on the language generation prefix.

**Strength And Weaknesses:**

Strong points
+ The idea looks very interesting to me.

Weak points
- Some descriptions are not very clear. For instance,

       - what are the prefix in the language models? How different prefix make impacts for the final results?

       - In Table-1, there are some classes (boat, chair, etc) that Imaginary perform worse than CLIP, it lacks explanations. Are these due to the image synthesis errors?

- The idea looks a bit straightforward, but the formulation is a bitter weak. It does not take the language and text-to-image generation errors into considerations. It would be perfect to have a GAN like loop between detectors and generators.

- For illustration part, I would suggest the author to show some cases why ImaginaryNet does not works but WSOD works, so that possible new ideas could be risen from analysis.

**Summary Of The Paper:**

This paper presents a method to learn object detectors from text-to-image synthesis data. The idea looks interesting to the community.
It first generates sentence description with prefix constraints and GPT-2, then generates imaginary images with DALLE-mini.
A WSOD is applied on sythensis image. The whole method is named ISOD. Experimental on PASCAL VOC 2007 shows that ISOD achieves close performance to WSOD on real images and annotations, while the performance could be even boosted when gradually increases real images and annotations.

**Summary Of The Review:**

This paper could be viewed as WSOD on image-to-text synthesis datasets. It needs to handle the domain differences between generated images and real images. The idea is overall novel, but the formulation is a bitter weak and the studies also not that comprehensive from my perspective.

I think the idea deserves a top conference, but the paper needs thorough studies to before it could be published.
I may change my rating based on the rebuttal.


==============================
The authors give detailed response to all my concerns. I raise the rating from 5=>6.
However, I would seriously suggest the authors make a revision to reformat the story of the work to move some important contents in appendix into the main body.

---

> ### Author Response · Authors · 2022-11-18
> **Authors Response - 1/3**
>
> ## Q1: Prefix in the Language Models
>
> Prefix is the initial input, which can provide important cues and the language model will extend the prefix to a longer sentence. Prefix is used in many zero-shot language generation or vision-and-language works [1,2,3]. The prefix will provide important cues for language models.
> Different prefixes will change the results of the language model and the results of the text-to-image synthesis model then as well. It will also affect the final performance of detection.
>
> Many works [1,4,5] prove that some simple prefixes can help the multimodal model calculate vision-and-language information better. Following these previous works, we consider four representative prefixes as follow:
> 1. A photo of a {class}
> 2. A photograph of a {class}
> 3. An image of a {class}
> 4. A {class}
>
> We conduct the comparison on different prefixes. Each prefix will generate 5K samples for training.
>
> |          | Aero  | Bick  | Bird  | Boat  | Bottle |  Bus  |  Car  |  Cat  | Chair |  Cow  | Table |  Dog  | Horse | Motor | Person | Plant | Sheep | Sofa  | Train |  TV   |  mAP  |
> | :------- | :---: | :---: | :---: | :---: | :----: | :---: | :---: | :---: | :---: | :---: | :---: | :---: | :---: | :---: | :----: | :---: | :---: | :---: | :---: | :---: | :---: |
> | Prefix 1 | 47.18 | 43.13 | 35.51 | 5.33  | 19.23  | 32.58 | 41.23 | 66.72 | 12.03 | 43.98 | 18.13 | 43.98 | 36.71 | 51.68 | 19.04  | 14.97 | 34.12 | 42.74 | 23.04 | 33.36 | 33.23 |
> | Prefix 2 | 40.77 | 48.30 | 35.59 | 7.33  | 23.73  | 45.56 | 38.24 | 45.30 | 13.83 | 35.68 | 15.73 | 38.39 | 41.82 | 42.24 | 12.95  | 16.47 | 39.64 | 37.34 | 24.57 | 42.48 | 32.30 |
> | Prefix 3 | 13.22 | 47.62 | 29.16 | 6.24  | 21.62  | 42.10 | 35.94 | 26.42 | 11.99 | 34.50 | 14.37 | 34.87 | 36.68 | 44.39 | 14.14  | 19.04 | 38.32 | 35.41 | 22.38 | 40.42 | 28.44 |
>
> For the Prefix (1) and (2), the language model can generate the sentence that describes the content of images based on the prefix. Meanwhile, the word in these prefixes, "photo" or "photograph", forces the text-to-image synthesis model to generate real-style images like photos. This is also the way for us to control the domain gap between the generated images and real images in the dataset.
>
> Prefix (3) obtains a much worse result. We can observe that although the language model still can generate the sentence describing objects well, the text-to-image synthesis model generates many images of cartoon or art style. This leads to a domain gap that makes the model get worse performance. Some classes that have a significant difference in real style and art style, such as cat, drop by a huge margin.
>
> Prefix (4) is an even worse prefix than (3). Most sentences generated by the language model are not describing the object, e.g., “A person should seek professional assistance to obtain an accommodation.”. We did not train our model based on the prefix (4) because it is hard for the text-to-image synthesis model to generate an image based on such a sentence without any object describing.

---

> ### Author Response · Authors · 2022-11-18
> **Authors Response - 2/3**
>
> ## Q2: Language and Text-to-image Generation Errors
>
> Thank you for your advice. To explore errors from language and text-to-image generation, we set up a CLIP-based filter, which measures the quality of generated data. Following the method of CLIP classification [1], the model will find out the most matched text in the group {“A photo of aeroplane”, “A photo of bicycle”, “A photo of bird”, …} with each image to classify the image. If the class of the image classified by CLIP is different from its initial class for generation, we consider this an error sample. We calculate the error rate on 25K generated data.
>
> |           | Aero | Bick | Bird |	Boat | Bottle | Bus | Car | Cat | Chair | Cow | Table | Dog | Horse | Motor | Person | Plant | Sheep | Sofa | Train | TV  |  All  |
> |   :---    | :---:| :---:| :---:| :---: |  :---: |:---:|:---:|:---:| :---: |:---:| :---: |:---:| :---: | :---: | :---:  | :---: | :---: |:---: | :---: |:---:| :---: |
> | Error Num |  35  | 82   | 147  | 87    |  112   | 310 |  115|  95 |  219  |  163|   87  | 160 |   117 |   116 |   451  |   53  |   114 |  140 |  155  | 689 |  3447 |
> |Error Ratio| 2.80 | 6.56 | 11.76| 6.96  |  8.96  |24.80| 9.20| 7.60| 17.52 |13.04| 6.96  |12.80|  9.36 |  9.28 |  36.08 | 4.24  |  9.12 | 11.20|  12.40|55.12| 13.79 |
>
> As the table above, we find that we have a 13.79% error rate overall and each class has a different error rate. This raises a question: How does generation error affect ImaginaryNet and can a lower error rate bring better performance?
> To answer this question, we randomly sample 5K correct samples and re-train detectors in ISOD.
>
> |                             | Aero  | Bick  | Bird  | Boat  | Bottle |  Bus  |  Car  |  Cat  | Chair |  Cow  | Table |  Dog  | Horse | Motor | Person | Plant | Sheep | Sofa  | Train |  TV   |  mAP  |
> | :-------------------------- | :---: | :---: | :---: | :---: | :----: | :---: | :---: | :---: | :---: | :---: | :---: | :---: | :---: | :---: | :----: | :---: | :---: | :---: | :---: | :---: | :---: |
> | ImaginaryNet                | 47.18 | 43.13 | 35.51 | 5.33  | 19.23  | 32.58 | 41.23 | 66.72 | 12.03 | 43.98 | 18.13 | 43.98 | 36.71 | 51.68 | 19.04  | 14.97 | 34.12 | 42.74 | 23.04 | 33.36 | 33.23 |
> | ImaginaryNet + Error Filter | 48.27 | 34.42 | 42.84 | 16.14 | 18.93  | 45.77 | 44.57 | 60.79 | 12.60 | 39.79 | 20.33 | 52.05 | 50.38 | 50.53 | 22.45  | 15.95 | 38.66 | 41.94 | 31.93 | 20.25 | 35.43 |
>
> We surprisedly find the performance of the model gains significantly the model with a large margin. We explained this as the error data will affect training and the model can learn better detector after filtering generation errors.
>
> ## Q3: GAN like Loop Between Detectors and Generators
>
> Thank you for this fantastic advice. We setup a GAN like loop by using the detector to find out the samples generated which have the highest loss in a group of samples. We train ImaginaryNet on 10K data first to initialize it and then continue to optimize it with the loop mentioned above on 5K extra samples. For each loop step, the generator will generate 5 samples, and the detector will choose one sample with the highest loss. To make a fair comparison, we use an ImaginaryNet trained with 15K data as the baseline.
>
> |                     | Aero  | Bick  | Bird  | Boat  | Bottle |  Bus  |  Car  |  Cat  | Chair |  Cow  | Table |  Dog  | Horse | Motor | Person | Plant | Sheep | Sofa  | Train |  TV   |  mAP  |
> | :------------------ | :---: | :---: | :---: | :---: | :----: | :---: | :---: | :---: | :---: | :---: | :---: | :---: | :---: | :---: | :----: | :---: | :---: | :---: | :---: | :---: | :---: |
> | ImaginaryNet        | 49.25 | 38.94 | 44.23 | 16.86 | 23.95  | 36.01 | 41.72 | 45.01 | 13.62 | 42.15 | 21.54 | 53.59 | 48.23 | 49.97 | 22.56  | 18.35 | 48.45 | 46.68 | 27.85 | 10.42 | 34.97 |
> | ImaginaryNet + Loop | 46.78 | 43.80 | 43.65 | 12.56 | 24.75  | 55.04 | 42.33 | 36.58 | 15.54 | 44.51 | 23.76 | 52.32 | 53.84 | 50.42 | 23.94  | 20.26 | 48.16 | 47.23 | 28.67 | 30.95 | 37.25 |
>
> From the table above, we can observe a significant gain by taking into account this GAN like loop. We think the reason for the improvement is similar to the GAN-based network: the goal of the detector is to minimize the loss of the samples, but the goal of the generator is to generate the sample that makes the detector has the highest loss. These adversarial goals make the network more effective.
>
> Note that, it is hard to optimize the parameter of the text-to-image synthesis model because directly adding the adversarial loss to generative model will make it unstable. This can be explained by the training task of the text-to-image synthesis model is complex and it is hard to optimize them without their original training data. Fortunately, because the generator can generate samples infinitely, we can use the detector to select the highest loss sample in a group. This makes the GAN like loop possible.

---

> ### Author Response · Authors · 2022-11-18
> **Authors Response - 3/3**
>
> ## Q4: Handling Domain Gap
>
> With the incredible increase in text-to-image synthesis models, images that are very similar to real photos can be synthesized. Meanwhile, we also use the prefix (see Q1 for details) as a tool to make the generated images as similar as possible to the real images. Some works [6,7] also prove the feasibility of using prompt in CLIP or ViT to do domain adaption. The significant improvement compared with those that do not use such prefix shows the importance to decrease the domain gap between generated images and real images.
>
> ## Q5: Classes That ImaginaryNet Perform Worse Than CLIP
>
> There are three classes, boat, chair, and table, that ImaginaryNet perform worse than CLIP.
>
> For class “boat”, we find that boats in most images generated are too small, and almost only water can be observed in the image. This makes the detector hard to learn this class.
>
> For class “chair”, we find that ImaginaryNet tends to generate chairs without any objects overlapping. However, chairs in the VOC test split are usually hidden under the desk or hidden by other things. This makes generated data much easier than the image in the VOC test split.
>
> For class “table”, we find that ImaginaryNet tends to generate a side view of a clean table without too many things on the tabletop. However, tables in the VOC test split are usually covered with a lot of things. Although we generated high-quality image, the shooting angle and tabletop is different from the image in the VOC test split.
>
> Different from CLIP, which is trained with many real images, the detector head in ImaginaryNet is only trained with these generated images. Therefore, such faults or differences may explain the inferior performance of ImaginaryNet against CLIP in these three classes.
>
> ## Q6: Cases That ImaginaryNet Does Not Works but WSOD Works
>
> Thank you for your advice. We add extra visualization in the revised version. We can observe that WSOD can better detect objects that are usually accompanying with each other, e.g., rider and motorbike. This is because we cannot know all the categories appearing in the generation image precisely. For example, we cannot know whether the generated images contain only a motorbike or a human riding a motorbike. If two objects are not accompanying usually, this may not influence the final results because of plenty of training samples. However, for objects accompanying usually, it is easy for the model to consider them as a whole object. This may motivate us to explore how to control the accompanying in the generation or better learn accompanying objects. A modified cross-attentive module like [8] for focusing on the key object described in the text prompt may help the model to decrease the possibilities of generating accompanying objects.
>
> Thank you for your valuable advice. All extra studies and discussions above will be supplemented to the revised version. Please let us know if you have any further concerns or advice.
>
> [1] Radford, Alec, et al. "Learning transferable visual models from natural language supervision." International Conference on Machine Learning. PMLR, 2021.
>
> [2] Wang, Zirui, et al. "Simvlm: Simple visual language model pretraining with weak supervision." arXiv preprint arXiv:2108.10904 (2021).
>
> [3] Cho, Jaemin, et al. "Unifying vision-and-language tasks via text generation." International Conference on Machine Learning. PMLR, 2021.
>
> [4] Ma, Teli, et al. "A simple long-tailed recognition baseline via vision-language model." arXiv preprint arXiv:2111.14745 (2021).
>
> [5] Xu, Mengde, et al. "A Simple Baseline for Open-Vocabulary Semantic Segmentation with Pre-trained Vision-Language Model." European Conference on Computer Vision. Springer, Cham, 2022.
>
> [6] Ge, Chunjiang, et al. "Domain Adaptation via Prompt Learning." arXiv preprint arXiv:2202.06687 (2022).
>
> [7] Zheng, Zangwei, et al. "Prompt Vision Transformer for Domain Generalization." arXiv preprint arXiv:2208.08914 (2022).
>
> [8] https://openreview.net/pdf?id=PUIqjT4rzq7

---

> ### Author Response · Authors · 2022-11-22
> **Is there any remaining concerns about our paper?**
>
> Thanks again for your comments! Is there any remaining concerns about our paper? We are more than delighted to address any concerns/questions you may have.

---

> > ### Comment · Reviewer_va3v · 2022-12-08
> > **Thanks for the detailed response**
> >
> > Many thanks for the authors' detailed response to my concerns.
> > It answers all my questions.
> > However, it may be not good that the author just put those response in the appendix in the revision.
> > I would suggest the authors make a revision to reformat the story of the work.
> >
> > As a result, I just raise my rating from 5 to 6.

---

> > > ### Author Response · Authors · 2022-12-13
> > > **Thank you!**
> > >
> > > Thanks for your thorough review and comments and your advice helps our paper be much stronger! Some of the discussions have been merged into the main body and we leave some contents in the appendix due to the 9-page limitation in the previously revised version. We will further reformat our paper to a better shape and move all important contents into the main body in the camera-ready version.
> > >
> > > Thank you again!
> > >
> > > Authors

---

> ### Author Response · Authors · 2022-12-01
> **We are very looking forward to discussing**
>
> Thank you again for your reviewing. Do you have any remaining concerns about our paper or response? We are very looking forward to discussing with you.

---

### Official Review · Reviewer_DNcv · 2022-10-23

**Confidence:** 5
**Correctness:** 3
**Technical Novelty And Significance:** 3
**Empirical Novelty And Significance:** 3
**Recommendation:** 8

**Clarity, Quality, Novelty And Reproducibility:**

The paper is around %10 of the papers in terms of clarity, quality, novelty and reproducibility.

**Strength And Weaknesses:**

Strengths:
+ Annotation is a labor-intensive task and mechanisms for reducing the cost of this process are very valuable.

+ A novel paradigm is introduced for using pre-trained text-to-image models for training object detectors.

+ Strong results over the baseline CLIP-based models.


Weaknesses:

1. My only concern with the paper is that it is lacking a comparison with other CLIP-based detection approaches introduced in Introduction.


Minor comments:

- "Humans can easily detect a known concept from language description without the demand of training in reality." => I would be very careful with such claims. I kind of disagree with this as there must have been some kind of (pre)training for humans to recognize the concept as known, either the language model, the other objects or the parts of the object/concept etc.

- First paragraph of the Intro: "RPNs or RoI heads" => Acronyms are used without introduction.
- "which is ”A photo of a”" => The opening quotation is incorrect. You can use `` in Latex to do this.
- "image encoder consisted of multi-layers" => "image encoder consisting of multi-layers".
- "a RPN network" => "an RPN network".
- "IMAGINARYNET apply RoI Pooling" => "IMAGINARYNET applies RoI Pooling". Please cite the relevant paper here for RoI pooling.
- "IMAGINARYNET obtain" => "IMAGINARYNET obtains".
- "IMAGINARYNET sample" => "IMAGINARYNET samples". The same problem recurs through out the text. Please check the paper carefully for more.
- "OICR(Tang et al., 2017)," => "OICR (Tang et al., 2017),".
- "and the structure of them are shown in Fig (2)." => "with the structure shown in Fig (2)."
- Eq 2, 3: dot should be placed after the eqs.
- "the (Tang et al., 2017)," => "Tang et al. (2017),". You can use a different cite command to get Tang et al. (2017).
- "for reaching no real images and manual annotations" => "for obtaining no real images and manual annotations".
- "the Edge Boxes algorithm" => Provide citation.

**Summary Of The Paper:**

In this paper, the authors study the problem of training an object detector without any real images or annotations. For this end, they propose generating synthetic (realistic) images using a pre-trained text-to-image model conditioned on words for the object categories as interest, and following a weakly-supervised object detection pipeline to train a detector given the generated image and the class labeled that conditioned it. The authors evaluate their approach on PASCAL VOC (and partially on MS COCO) with a comparison to other pre-training based object detectors, weakly-supervised detectors and a supervised method.


**Summary Of The Review:**

Novel method, strong results. Missing important comparisons to other CLIP-based approaches.

---

> ### Author Response · Authors · 2022-11-18
> **Authors Response**
>
> ## Q1: Comparison with Other CLIP-based Detection Approaches
>
> We add the extra comparison with these CLIP-based Detection approaches. Due to these models, e.g., RegionCLIP [1], ViLD [2], and Detic [3], use different human-annotated real detection data, we adopt the labeled dataset for training each model. For a fair comparison, we adopt ImaginaryNet(SSOD) for leveraging both imaginary data and the same labeled data. We compare results on 20 base class in VOC 2007 test split. For ViLD, we directly provide the results from their paper.
>
> |                     |   Annotated Real Data   | Unannotated Real Data | Unannotated Imaginary Data |  mAP  |
> | :------------------ | :---------------------: | :-------------------: | :------------------------: | :---: |
> | RegionCLIP          |       74K COCO-48       |        1.5M CC        |         $\emptyset$        | 68.35 |
> | Detic               |  165K COCO + 100K LVIS  |     14M ImageNet      |         $\emptyset$        | 79.44 |
> | ViLD                | 5K VOC 2007 + 100K LVIS |      $\emptyset$      |         $\emptyset$        | 78.9  |
> | ImaginaryNet(SSOD) |       5K VOC 2007       |      $\emptyset$      |        5K Imaginary        | 80.30 |
> | ImaginaryNet(SSOD) |       5K VOC 2007       |      $\emptyset$      |       10K Imaginary        | 80.59 |
> | ImaginaryNet(SSOD) |       5K VOC 2007       |     10K VOC 2012      |       10K Imaginary        | 82.00 |
>
> From the table, we can observe ImaginaryNet(SSOD) still obtain the best performance. This proves the effectiveness of generated data from ImaginaryNet.
>
> ## Q2: Minor comments
>
> Thank you for your careful review. We will discuss them one by one.
>
> ### Humans can easily detect a known concept from language description without the demand of training in reality.
>
> Yes, we agree that even humans must have some kind of knowledge to recognize the concept. We have rewritten this sentence to “Without the demand of training in reality, humans are able of detecting a new category of object simply based on the language description on its visual characteristics.”
>
> ### Acronyms are used without introduction
>
> We have added the explanation in the revised version.
>
> ### Opening quotation, equation, and citation style
>
> Thank you for your advice! We have changed these in the revised version.
>
> ### Citation of Edge Boxes
>
> We have fixed the lack of citations in the revised version.
>
> ### Other typos and writing problems
>
> We have carefully checked the paper and fixed them.
>
> Thank you for your careful and thorough review. We will polish our paper to a better shape in the revised version. We are glad to response your further concerns or advice.
>
> [1] Zhong, Yiwu, et al. "Regionclip: Region-based language-image pretraining." Proceedings of the IEEE/CVF Conference on Computer Vision and Pattern Recognition. 2022.
>
> [2] Gu, Xiuye, et al. "Open-vocabulary object detection via vision and language knowledge distillation." arXiv preprint arXiv:2104.13921 (2021).
>
> [3] Zhou, Xingyi, et al. "Detecting twenty-thousand classes using image-level supervision." European Conference on Computer Vision. Springer, Cham, 2022.

---

> > ### Comment · Reviewer_DNcv · 2022-11-22
> > **Re: Authors response**
> >
> > Dear authors,
> >
> > Thank you for the responses, especially the additional experiments. I don't have any further questions/comments.
> >
> > Best

---

> > > ### Author Response · Authors · 2022-11-24
> > > **Thank you!**
> > >
> > > We are very appreciated for your thorough review and helping the paper into a better shape. We will continue polishing our paper.
> > >
> > > Thank you again!
> > >
> > > Authors

---

> ### Author Response · Authors · 2022-11-22
> **Is there any remaining concerns about our paper?**
>
> Thanks again for your comments! Is there any remaining concerns about our paper? We are more than delighted to address any concerns/questions you may have.

---

### Official Review · Reviewer_n5tS · 2022-10-24

**Confidence:** 3
**Correctness:** 4
**Technical Novelty And Significance:** 2
**Empirical Novelty And Significance:** Not applicable
**Recommendation:** 6

**Clarity, Quality, Novelty And Reproducibility:**

The quality of the writing is good.  The paper does suffer from low novelty since others have explored similar ideas to train deep learning models.  The engineering apsects of the work are clear; however, I couldn't get an intuitive feel for why the paper outperforms a method such as CLIP.  I feel that the paper will benefit from including a summary of how this work is different from prior work and what aspects of the model leads to better performance.  Is it training?  Is it because the model has a lot more capacity than some other models? Is it because we now have models to generate highly realistic visual data?

**Strength And Weaknesses:**

The approach presented herein is well-motivated; however, given prior works that also advocate the use of synthetic data for training deep learning model, the novelty-aspect of this work is somewhat low.  Section 2.2, for example, lists some of these works.  Reading this section it seems that existing schemes are not as robust (or useful) for model training simply because the quality, i.e., visual realism, of the generated images do not match "real" images.  This subsequently leads to a domain gap, which must be addressed by using manually-annotated real images.  If this is true then one may reasonable assume that with our increased ability to generated visual realistic images, some of the existing methods may work as well as the approach developed in this work.  I feel that the paper needs to do a better job of distinguishing itself from prior art.

Along the same lines, while the work shows that ImaginaryNet outperfroms CLIP, it is not immediately obvious to me why is it so.  Posing this question another way, do you think CLIP can achieve comparable performance if allowed to train on larger training data (generated data, of course).


**Summary Of The Paper:**

The paper proposes a scheme for training object detection networks using self-generated synthetic data.  The system presented herein does not require manually-annotated real images, which are hard to come by.  Rather, the system leverages the spectacular advances in text-to-image generation system to construct visually realistic data that serves the role of traditional object detection datasets.  The proposed scheme is able to benefit from the "real" data when available.  The paper also includes a set of experiments that confirm the useful of this approach.

**Summary Of The Review:**

The work is interesting and it is moves the field of deep learning in the right direction.  The approach presented here reduces our reliance on manually annotated real data for the task of object detection, which is highly desireable.  At the same time it is somewhat difficult to ascertain what makes this model better than other approaches that also use generated data.  The novelty, I feel, is on the lower-end of the spectrum.

---

> ### Author Response · Authors · 2022-11-18
> **Authors Response**
>
> ## Q1: Summary of the Difference from Prior Work
>
> Sim2real aims to use the simulator engines to simulate images for the model. Some works [1,2,3] use engines to simulate data for different vision tasks, such as emotion classification. In object detection, [4,5,6] use simulated images with 2D or 3D engine, e.g., UE4, for industrial object detection. Because of the large domain gap caused by limitation of simulating engine, Sim2real and subsequent domain adaption methods focus on reducing domain gap. However, with the progress in text-to-image synthesis, domain gap of images generated by such models has been largely reduced. The key problem is how to effectively employ text-to-image synthesis model for generating diverse images with proper content and quality. To this end, we propose ImaginaryNet, which can even archive 70% performance in ISOD compared with the weakly supervised counterpart of the same backbone.
>
> ## Q2: Reason that ImaginaryNet Outperforms CLIP
>
> We think that CLIP cannot use the information from the overlapping proposals because its pre-training is based on image-text matching. But the detection head of ImaginaryNet is designed for the training detection task. We conduct the extra comparison by fine-tuning the CLIP model on the same data we used for ImaginaryNet.
>
> |                 | Aero  | Bick  | Bird  | Boat  | Bottle |  Bus  |  Car  |  Cat  | Chair |  Cow  | Table |  Dog  | Horse | Motor | Person | Plant | Sheep | Sofa  | Train |  TV   |  mAP  |
> | :-------------- | :---: | :---: | :---: | :---: | :----: | :---: | :---: | :---: | :---: | :---: | :---: | :---: | :---: | :---: | :----: | :---: | :---: | :---: | :---: | :---: | :---: |
> | CLIP            | 16.30 | 17.15 | 19.42 | 8.80  | 14.53  | 30.28 | 25.83 | 25.92 | 20.34 | 27.10 | 21.85 | 26.00 | 20.32 | 22.59 | 10.09  | 13.43 | 26.24 | 27.82 | 19.26 | 23.59 | 20.84 |
> | CLIP + Finetune | 18.74 | 20.37 | 15.72 | 5.10  |  9.78  | 32.86 | 35.50 | 27.98 | 12.19 | 23.79 | 12.69 | 41.42 | 29.17 | 14.91 | 24.07  | 6.29  | 34.89 | 29.02 | 21.66 | 24.04 | 22.01 |
> | ImaginaryNet    | 47.18 | 43.13 | 35.51 | 5.33  | 19.23  | 32.58 | 41.23 | 66.72 | 12.03 | 43.98 | 18.13 | 43.98 | 36.71 | 51.68 | 19.04  | 14.97 | 34.12 | 42.74 | 23.04 | 33.36 | 33.23 |
>
> As shown in the above table, ImaginaryNet still outperforms CLIP even after fine-tuning CLIP on our generated data. But CLIP is still a strong baseline for this task because it uses a large-scale pre-training and can be used to make a comparison in ISOD, the strict setting that models cannot access any real images and annotations.
>
> Thank you for your constructive comments. We will add discussions above to the revised version. If you have any further concerns or advice, we are looking forward to discussing with you.
>
> [1] Akhyani, Saba, et al. "Towards inclusive hri: Using sim2real to address underrepresentation in emotion expression recognition." arXiv preprint arXiv:2208.07472 (2022).
>
> [2] Sadeghi, Fereshteh, and Sergey Levine. "Cad2rl: Real single-image flight without a single real image." arXiv preprint arXiv:1611.04201 (2016).
>
> [3] Wang, Yu-Xiong, et al. "Low-shot learning from imaginary data." Proceedings of the IEEE conference on computer vision and pattern recognition. 2018.
>
> [4] Borrego, Joao, et al. "A generic visual perception domain randomisation framework for Gazebo." 2018 IEEE International Conference on Autonomous Robot Systems and Competitions (ICARSC). IEEE, 2018.
>
> [5] Tremblay, Jonathan, et al. "Training deep networks with synthetic data: Bridging the reality gap by domain randomization." Proceedings of the IEEE conference on computer vision and pattern recognition workshops. 2018.
>
> [6] Horváth, Dániel, et al. "Object detection using sim2real domain randomization for robotic applications." IEEE Transactions on Robotics (2022).

---

> ### Author Response · Authors · 2022-11-22
> **Is there any remaining concerns about our paper?**
>
> Thanks again for your comments! Is there any remaining concerns about our paper? We are more than delighted to address any concerns/questions you may have.

---

> ### Author Response · Authors · 2022-12-01
> **We are very looking forward to discussing**
>
> Thank you again for your reviewing. Do you have any remaining concerns about our paper or response? We are very looking forward to discussing with you.

---

### Official Review · Reviewer_w1NM · 2022-10-25

**Confidence:** 4
**Correctness:** 3
**Technical Novelty And Significance:** 3
**Empirical Novelty And Significance:** 3
**Recommendation:** 6

**Clarity, Quality, Novelty And Reproducibility:**

The overall quality of the paper is acceptable. More clarity is needed in the experiments and ablation studies. The overall ideas of the work are original, however, the framework is developed by putting together existing models in a pipeline, e.g. language model via GPT-2, text-to-image synthesis model via DALLE-mini, and proposal generator via selective search.
Based on what is described in the paper, the proposed framework should be reproducible.

**Strength And Weaknesses:**

Strength:
1. Using language and synthesis models is a clever idea to improve the existing WSOD algorithm.
2. Extensive experimental results to demonstrate the strength of the proposed method

Weakness:
1. The effect of the limitation caused by synthesis models was not well studied and discussed although there is an ablation study on the effectiveness of the language model.
2. The experiments and ablation studies sections are not well organized and hard to follow. There is some confusion in the settings, e.g. ImaginaryNet ISOD, WSOD, SSOD.


**Summary Of The Paper:**

This paper introduces a framework, named ImaginaryNet, to train object detectors using images generated by a text-to-image model and class labels from a pre-trained language model.
This is also a learning scheme, named Imaginary-Supervised Object Detection, for training object detectors without requiring real images and manual annotations. This scheme is based on a weakly-supervised object detection (WSOD) algorithm, i.e. OICR, with the object proposal and class labels obtained from ImaginaryNet.
The paper's main contribution is in using the language model and synthesis model to build a learning scheme on top of the existing WSOD scheme without requiring real images or manual annotations.


**Summary Of The Review:**

My initial rating is a weak reject. The main reason for rejection is that the paper did not put enough focus on the synthesis model. This makes the paper's claim of learning object detectors without real images weaker since there could be an object class that the synthesis model cannot generate without learning it first. In that case, we still need to have real images. For example, learning a "human hand" detector would require the synthesis model to generate an image with a "human hand".

---

> ### Author Response · Authors · 2022-11-18
> **Authors Response - 1/2**
>
> ## Q1: Limitation Discussion and Further Ablation Study
>
> Thank you for your advice. We further discuss the limitations and conduct extra ablation studies on the quantity and quality of the generated images.
>
> ### Infinite generation may not bring in higher performance when the number is enough.
>
> We extend Tab. 4, on Page. 8 of the original paper, to explore what will happen if further increasing the number of generation samples.
>
> |      | Aero  | Bick  | Bird  | Boat  | Bottle |  Bus  |  Car  |  Cat  | Chair |  Cow  | Table |  Dog  | Horse | Motor | Person | Plant | Sheep | Sofa  | Train |  TV   |  mAP  |
> | :--- | :---: | :---: | :---: | :---: | :----: | :---: | :---: | :---: | :---: | :---: | :---: | :---: | :---: | :---: | :----: | :---: | :---: | :---: | :---: | :---: | :---: |
> | 2K   | 39.35 | 43.68 | 33.36 | 4.51  | 13.84  | 29.39 | 38.71 | 59.48 | 10.69 | 35.75 | 21.06 | 37.43 | 28.97 | 43.64 | 17.20  | 13.55 | 33.10 | 35.71 | 17.51 | 29.73 | 29.33 |
> | 3K   | 39.15 | 40.10 | 33.18 | 3.40  | 17.19  | 32.34 | 37.21 | 53.69 | 11.50 | 41.29 | 19.75 | 39.34 | 35.24 | 48.15 | 16.30  | 13.74 | 29.83 | 42.62 | 25.38 | 28.01 | 30.37 |
> | 4K   | 41.92 | 41.99 | 35.64 | 4.71  | 17.88  | 35.88 | 38.28 | 58.10 | 11.06 | 35.17 | 24.19 | 33.36 | 36.34 | 45.88 | 17.80  | 14.30 | 30.71 | 41.97 | 26.45 | 40.82 | 31.62 |
> | 5K   | 47.18 | 43.13 | 35.51 | 5.33  | 19.23  | 32.58 | 41.23 | 66.72 | 12.03 | 43.98 | 18.13 | 43.98 | 36.71 | 51.68 | 19.04  | 14.97 | 34.12 | 42.74 | 23.04 | 33.36 | 33.23 |
> | 10K  | 45.10 | 51.31 | 34.22 | 5.34  | 27.65  | 36.62 | 39.35 | 62.00 | 13.41 | 45.31 | 20.45 | 41.41 | 51.13 | 52.25 | 21.32  | 21.48 | 35.20 | 44.48 | 22.06 | 33.82 | 35.20 |
> | 15K  | 49.25 | 38.94 | 44.23 | 16.86 | 23.95  | 36.01 | 41.72 | 45.01 | 13.62 | 42.15 | 21.54 | 53.59 | 48.23 | 49.97 | 22.56  | 18.35 | 48.45 | 46.68 | 27.85 | 10.42 | 34.97 |
> | 20K  | 48.44 | 41.09 | 43.20 | 16.29 | 25.04  | 40.14 | 43.49 | 43.83 | 13.97 | 40.10 | 25.96 | 49.82 | 51.14 | 51.85 | 22.68  | 17.46 | 44.08 | 48.45 | 31.02 | 18.28 | 35.82 |
> | 25K  | 49.11 | 41.34 | 46.78 | 17.02 | 21.09  | 33.98 | 41.42 | 61.53 | 14.43 | 43.63 | 22.27 | 51.88 | 55.27 | 51.45 | 21.54  | 16.47 | 46.63 | 46.51 | 26.06 | 17.21 | 36.28 |
>
> From the above table, one can see that ImaginaryNet obtains the mAP of 36.28 when using 25K generated samples. Nonetheless, the growth of performance has slowed down when the number of generated samples is more than 10K. Thus, we speculate that 36.28 approaches the upper bound of the performance. We note that it may be hard for our method to surpass the method training on full annotated real images. How to utilize the generative models effectively for refreshing this upper bound will be a question in the future study.
>
> ### Generation quality will limit performance.
>
> Generative quality will affect the final performance. We discuss the generative quality from two aspects: (1) diversity and photo-reality, and (2) error rates.
>
> The diversity of images generated will help the model learn more knowledge and improve its robustness. Diversity is affected by not only the text-to-image synthesis model but also language prompts. This is also the motivation that we introduce the language model to the framework to obtain better diversity. Meanwhile, higher photo-reality also helps the model to obtain a better performance due to low domain gap to real images. With the incredible increase in text-to-image synthesis models, images that are very similar to real photos can be synthesized. Therefore, we believe that we can obtain images generated with higher and higher photo-reality.
>
> On the other hand, the language and text-to-image synthesis model may bring in errors during their generation process. To explore errors from language and text-to-image generation, we set up a CLIP-based filter, which measures the quality of generated data. Following the method of CLIP classification [1], the model will find out the most matched text in the group {“A photo of aeroplane”, “A photo of bicycle”, “A photo of bird”, …} with each image to do classification. If the class of the image classified by CLIP is different from its initial class used for generation, we consider this an error sample. We calculate the error rate on 25K generated data.
>
> (Continued in Authors Response 2/2)

---

> > ### Comment · Reviewer_w1NM · 2022-12-08
> > **RE: Authors Response - 1/2**
> >
> > Thank you for your response. I have no further comments on this question.

---

> ### Author Response · Authors · 2022-11-18
> **Authors Response - 2/2**
>
> (From the Authors Response 1/2)
>
> |           | Aero | Bick | Bird |	Boat | Bottle | Bus | Car | Cat | Chair | Cow | Table | Dog | Horse | Motor | Person | Plant | Sheep | Sofa | Train | TV  |  All  |
> |   :---    | :---:| :---:| :---:| :---: |  :---: |:---:|:---:|:---:| :---: |:---:| :---: |:---:| :---: | :---: | :---:  | :---: | :---: |:---: | :---: |:---:| :---: |
> | Error Num |  35  | 82   | 147  | 87    |  112   | 310 |  115|  95 |  219  |  163|   87  | 160 |   117 |   116 |   451  |   53  |   114 |  140 |  155  | 689 |  3447 |
> |Error Ratio| 2.80 | 6.56 | 11.76| 6.96  |  8.96  |24.80| 9.20| 7.60| 17.52 |13.04| 6.96  |12.80|  9.36 |  9.28 |  36.08 | 4.24  |  9.12 | 11.20|  12.40|55.12| 13.79 |
>
> As the table above, we find that we have a 13.79% error rate overall and each class has a different error rate. This raises a question: How does generation error affect ImaginaryNet and can a lower error rate bring better performance?
>
> To answer this question, we randomly sample 5K correct samples and re-train detectors in ISOD.
>
> |                             | Aero  | Bick  | Bird  | Boat  | Bottle |  Bus  |  Car  |  Cat  | Chair |  Cow  | Table |  Dog  | Horse | Motor | Person | Plant | Sheep | Sofa  | Train |  TV   |  mAP  |
> | :-------------------------- | :---: | :---: | :---: | :---: | :----: | :---: | :---: | :---: | :---: | :---: | :---: | :---: | :---: | :---: | :----: | :---: | :---: | :---: | :---: | :---: | :---: |
> | ImaginaryNet                | 47.18 | 43.13 | 35.51 | 5.33  | 19.23  | 32.58 | 41.23 | 66.72 | 12.03 | 43.98 | 18.13 | 43.98 | 36.71 | 51.68 | 19.04  | 14.97 | 34.12 | 42.74 | 23.04 | 33.36 | 33.23 |
> | ImaginaryNet + Error Filter | 48.27 | 34.42 | 42.84 | 16.14 | 18.93  | 45.77 | 44.57 | 60.79 | 12.60 | 39.79 | 20.33 | 52.05 | 50.38 | 50.53 | 22.45  | 15.95 | 38.66 | 41.94 | 31.93 | 20.25 | 35.43 |
>
> We surprisedly find the performance of the model gains significantly and obtain the performance of the model trained by 20K un-filtered data. We explained this as the error data will affect training and the model can learn better detector after filtering generation errors.
>
> ## Q2: Confusion of Settings
>
> Sorry for the confusion. ImaginaryNet can be viewed as a combination of a detection data generator and an object detector module, which can work with or without real images for learning the object detector. To answer the question: can we learn object detectors without real images and annotations, we set up a novel detection setting: Imaginary-Supervised Object Detection (ISOD). In ISOD, no real images and manual annotations can be used for training object detection. Experiments show that ImaginaryNet in ISOD setting can obtain about 70% performance compared with the weakly supervised counterpart of the same backbone trained on real data. We also introduce ImaginaryNet to two common object detection settings, WSOD and SSOD, where annotations on real images are incomplete. In WSOD, real images exist but only class-level annotations can be accessed. In SSOD, only part of the images has annotations. Experiments show ImaginaryNet significantly improves the baseline while achieving state-of-the-art or comparable performance by incorporating real images and manual annotations.
>
> ## Q3: Synthesis Model Cannot Generate Without Learning It First
>
> Different from class-to-image synthesis models, text-to-image synthesis models have a much stronger capability of composition  and extrapolation [2,3,4]. For example, the model may be trained with only “black cats” and “white swans”, but it can generate “black swans”. Meanwhile, text-to-image synthesis models can even create new objects with detailed descriptions. For example, they can generate the image with the description “an elephant-like creature with two wings.” We provide examples in the appendix.
>
> Thank you for your valuable comments. All discussions above will be included in the revised version. If you have any further concerns or advice, please let us know.
>
> [1] Radford, Alec, et al. "Learning transferable visual models from natural language supervision." International Conference on Machine Learning. PMLR, 2021.
>
> [2] Ramesh, Aditya, et al. "Hierarchical text-conditional image generation with clip latents." arXiv preprint arXiv:2204.06125 (2022).
>
> [3] Saharia, Chitwan, et al. "Photorealistic Text-to-Image Diffusion Models with Deep Language Understanding." arXiv preprint arXiv:2205.11487 (2022).
>
> [4] Rombach, Robin, et al. "High-resolution image synthesis with latent diffusion models." Proceedings of the IEEE/CVF Conference on Computer Vision and Pattern Recognition. 2022.

---

> > ### Comment · Reviewer_w1NM · 2022-12-08
> > **RE: Authors Response - 2/2**
> >
> > Thank you for your response. I still have some comments as follows.
> >
> > For Q2, please provide more clarification for ImaginaryNet being used in various object detection settings including ISOD, WSOD and SSOD in the text. I am still not very clear that the language model and text-to-image model were used in WSOD and SSOD settings.
> >
> > For Q3, only part of my question was addressed. Could your text-to-image synthesis model really generate "a photo of human hand" or "a photo of credit card"? or we have to describe those new objects using only learned classes/vocabularies/concept?

---

> > > ### Author Response · Authors · 2022-12-13
> > > **Further Explanation - 1/2**
> > >
> > > Thank you for your thorough comments! We will further explain these questions.
> > >
> > > ## Please provide more clarification for ImaginaryNet being used in various object detection settings including ISOD, WSOD and SSOD in the text.
> > >
> > > In all settings including ISOD, WSOD, and SSOD, ImaginaryNet is adopted to generate photo-realistic and diverse images of specific object categories by (1) generating the text description by language model based on the sampled class label, (2) generating the image by text-to-image synthesis model based on the generated text description. Then, we use selective search to generate a set of region proposals for each synthesis image.
> > >
> > > In ISOD, no real images and annotations can be accessed. ImaginaryNet will generate a dataset $D_w^G=\{(\mathbf{I}_w^G,\mathbb{P}^G,\mathbf{Y}_w^G)\}$. Here, $\mathbf{I}_w^G$ denotes an image generated by ImaginaryNet. $\mathbf{Y}_w^G=\left[y_1^G,\ldots,y_C^G\right]$ indicates the image-level weak-annotation, where $y_c^G\in\{0,1\}$ indicates the if $c$ is the initial class label for $\mathbf{I}_w^G$, and $C$ is the number of categories. $\mathbb{P}^G=\{\mathbf{b}_1,\ldots,\mathbf{b}_M\}$ is a set of region proposals from $\mathbf{I}_w^G$ obtained by the selective search algorithm, where $\mathbf{b}_i$ is the $i$-th proposal box defined by $\left[x_\mathrm{min},y_\mathrm{min},x_\mathrm{max},y_\mathrm{max}\right]$ that specifies its top-left corner $(x_\mathrm{min},y_\mathrm{min})$ and its bottom-right corner $(x_\mathrm{max},y_\mathrm{max})$. In the training phase of ISOD, we aim to learn the detection head to predict the classification probability of each proposal, i.e., $\hat{\mathbf{Y}}=f\left(\mathbf{I}_w^G,\mathbb{P}^G;\theta^\mathrm{ISOD}\right)$ only using the generated dataset $D_w^G=\{(\mathbf{I}_w^G,\mathbb{P}^G,\mathbf{Y}_w^G)\}$ following the OICR [1] method.
> > >
> > > In WSOD, we further have a real dataset $D_w^R=\{(\mathbf{I}_w^R,\mathbb{P}^R,\mathbf{Y}_w^R)\}$. The real image $\mathbf{I}_w^R$ and corresponding image-level weak-annotation $\mathbf{Y}_w^R=\left[y_1^R,...,y_C^R\right]$ can be accessed in this setting, where $y_c^R\in\{0,1\}$ indicates the presence of at least one instance of $c$-th category and $C$ is the number of categories. Also, ImaginaryNet will generate a dataset $D_w^G=\{(\mathbf{I}_w^G,\mathbb{P}^G,\mathbf{Y}_w^G)\}$. In the training phase of WSOD, we aim to learn the detection head to predict the classification probability of each proposal, i.e., $\hat{\mathbf{Y}}=f(\mathbf{I},\mathbb{P};\theta^\mathrm{WSOD})$, using generated dataset $D_w^G=\{(\mathbf{I}_w^G,\mathbb{P}^G,\mathbf{Y}_w^G)\}$ and real dataset $D_w^R=\{(\mathbf{I}_w^R,\mathbb{P}^R,\mathbf{Y}_w^R)\}$, where $\mathbf{I}$ represents $\mathbf{I}_w^G$ or $\mathbf{I}_w^R$ and $\mathbb{P}$ represents $\mathbb{P}^G$ or $\mathbb{P}^R$. In general, any WSOD methods can be adopted to learn $\hat{\mathbf{Y}}=f(\mathbf{I},\mathbb{P};\theta^\mathrm{WSOD})$, and we use OICR [1] as an example in our implementation.
> > >
> > > In SSOD, we have a real labeled dataset $D_f^R=\{(\mathbf{I}_f^R,\mathbf{Y}_f^R)\}$ and an un-labeled dataset $D_u^R=\{(\mathbf{I}_u^R)\}$. The real image $\mathbf{I}_f^R$, corresponding instance-level annotations $\mathbf{Y}_f^R=\left[(\mathbf{b}_1,c_1),\ldots\right]$ with $N^R$ elements, and unlabeled image $\mathbf{I}_u^R$ can be accessed. Here, $N^R$ is the number of object instances associated with $\mathbf{I}_f^R$, and $\mathbf{b}_i$ is the $i$-th object localization bounding box defined by $[x_\mathrm{min},y_\mathrm{min},x_\mathrm{max},y_\mathrm{max}]$ that specifies its top-left corner $(x_\mathrm{min},y_\mathrm{min})$ and its bottom-right corner $(x_\mathrm{max},y_\mathrm{max})$. Also, ImaginaryNet further generates a dataset $D_w^G=\{(\mathbf{I}_w^G,\mathbb{P}^G,\mathbf{Y}_w^G)\}$. For simplicity, we ignore $\mathbf{Y}_w^G$ or $\mathbb{P}^G$ and combine $\{(\mathbf{I}_u^R)\}$ and $\{(\mathbf{I}_w^G)\}$ to form the unlabeld dataset $D_u$ . The detection head to predict the classification probability of each proposal, $\hat{\mathbf{Y}}=f(\mathbf{I};\theta^\mathrm{SSOD})$, where $\mathbf{I}$ represents $\mathbf{I}_u^G$, $\mathbf{I}_f^R$ or $\mathbf{I}_u^R$. During the training of SSOD, we use the Unbiased Teacher [2] method to optimize the detection head $\theta^\mathrm{SSOD}$ with the combination of unlabeled dataset $D_u$ and real labeled dataset $D_f^R=\{(\mathbf{I}_f^R,\mathbf{Y}_f^R)\}$.

---

> > > ### Author Response · Authors · 2022-12-13
> > > **Further Explanation - 2/2**
> > >
> > > ## Could your text-to-image synthesis model really generate "a photo of human hand" or "a photo of credit card"? or we have to describe those new objects using only learned classes/vocabularies/concept?
> > >
> > > Yes, the text-to-image synthesis model can generate "a photo of human hand" or "a photo of credit card". We provide cases of "human hand" and "credit card" generated by their name in this anonymous link [ https://sites.google.com/view/imaginarynetrebuttal/ ]. We can notice the model successfully generate the right object with high quality.
> > >
> > > Can the text-to-image synthesis model even generate based on the description of their visual characteristics?
> > >
> > > To figure out this question, we provide the visual characteristics of "credit card" and a specific class of a bird from the CUB dataset instead of their class name in the prompt following the rules in ATTN-GAN [3]. For "credit card", we describe it with “A photo of blue card on the table with 1mm thickness and rounded corner, card numbers in the card with title "Bank of Card"”, and for the bird, we describe it with “A photo of bird has a green crown black primaries and a white belly”. We can find the vivid and attractive results generated, which are matching with the text description well. These cases prove the exciting ability of the text-to-image synthesis model. You can find these cases in this anonymous link [ https://sites.google.com/view/imaginarynetrebuttal/ ]. Please note that the CUB dataset is not used for training the text-to-image synthesis model.
> > >
> > > Why can the text-to-image synthesis model generate based on not only the name of these classes but also the description of their visual characteristics?
> > >
> > >  This is because the text encoder in the text-to-image synthesis model transforms the text prompt into the vector in semantic space. This allows the model to generate almost infinite classes.
> > >
> > > [1] Tang, Peng, et al. "Multiple instance detection network with online instance classifier refinement." Proceedings of the IEEE conference on computer vision and pattern recognition. 2017.
> > >
> > > [2] Liu, Yen-Cheng, et al. "Unbiased teacher for semi-supervised object detection." arXiv preprint arXiv:2102.09480 (2021).
> > >
> > > [3] Xu, Tao, et al. "Attngan: Fine-grained text to image generation with attentional generative adversarial networks." Proceedings of the IEEE conference on computer vision and pattern recognition. 2018.

---

> ### Author Response · Authors · 2022-11-22
> **Is there any remaining concerns about our paper?**
>
> Thanks again for your comments! Is there any remaining concerns about our paper? We are more than delighted to address any concerns/questions you may have.

---

> ### Author Response · Authors · 2022-12-01
> **We are very looking forward to discussing**
>
> Thank you again for your reviewing. Do you have any remaining concerns about our paper or response? We are very looking forward to discussing with you.

---

### Decision · Program_Chairs · 2023-01-20

**Decision:**

Accept: poster

**Justification For Why Not Higher Score:**

The paper's presentation (writing quality, clarity) could be greatly improved. Technically, the approach has limitations primarily tied to the generative model's capability. Generated images have been used to train detectors before, which limits the technical novelty.

**Justification For Why Not Lower Score:**

The proposed method achieves really strong performance compared to recent CLIP-based detectors. This makes it practically useful, while also being interesting.

**Metareview: Summary, Strengths And Weaknesses:**

*Summary*: The paper tackles the problem of training object detectors without directly using any annotations or real images. The authors use a text to image generative model to generate images, followed by training a weakly supervised detector. The model is evaluated on COCO and VOC.

*Strengths*: (1) The idea of using text to image generative models for training detectors is interesting. Since these models can be conditioned on object categories, they generate object-centric images. (2) Strong performance compared to recent models like Detic and ViLD.

*Weaknesses*: (1) The method's performance is bottlenecked by the generative model.  As the authors also note in their response, the performance saturates with many generated samples. (2) The experimental details should be better explained.

**Note From Pc:**

if the above contains the word "oral" or "spotlight" please see: "oral" presentation means -> notable-top-5% and "spotlight" means -> notable-top-25%. As stated in our emails, we are disassociating presentation type from AC recommendations